# Kilometre-scale simulations over Fennoscandia reveal a large loss of tundra due to climate warming

Fredrik Lagergren[1], Robert G. Björk[2, 3], Camilla Andersson[4], Danijel Belušić[4, 5], Mats P. Björkman[2, 3 now at 6], Erik Kjellström[4], Petter Lind[4], David Lindstedt[4], Tinja Olenius[4], Håkan Pleijel[6], Gunhild Rosqvist[7] and Paul A. Miller[1]

[1]Department of Physical Geography and Ecosystem Science, Lund University, Lund, 223 62, Sweden

[2]Department of Earth Sciences, University of Gothenburg, Gothenburg, 405 30, Sweden

[3]Gothenburg Global Biodiversity Centre, Gothenburg, 405 30, Sweden

[4]Swedish Meteorological and Hydrological Institute, Norrköping, 601 76, Sweden

[5]Department of Geophysics, Faculty of Science, University of Zagreb, Zagreb, 10 000, Croatia

[6]Department of Biological & Environmental Sciences, University of Gothenburg, Gothenburg, 405 30, Sweden

[7]Department of Physical Geography, Stockholm University, Stockholm, 106 91, Sweden

*Correspondence to*: Fredrik Lagergren (Fredrik.Lagergren@nateko.lu.se)

**Abstract.** The Fennoscandian boreal and mountain regions harbour a wide range of vegetation types, from boreal forest to high alpine tundra and barren soils. The area is facing a rise in air temperature above the global average and changes in temperature and precipitation patterns. This is expected to alter the Fennoscandian vegetation composition and change the conditions for areal land-use such as forestry, tourism and reindeer husbandry. In this study we used a unique high-resolution (3 km) climate scenario with considerable warming resulting from strongly increasing carbon dioxide emissions to investigate how climate change can alter the vegetation composition, biodiversity and availability of suitable reindeer forage. Using a dynamical vegetation model, including a new implementation of potential reindeer grazing, resulted in simulated vegetation maps of unprecedented high resolution for such a long time period and spatial extent. The results were evaluated at the local scale using vegetation inventories and for the whole area against satellite-based vegetation maps. A deeper analysis of vegetation shifts related to statistics of threatened species was performed in six "hotspot" areas containing records of rare and threatened species. In this high emission scenario, the simulations show dramatic shifts in the vegetation composition, accelerating at the end of the century. Alarmingly, the results suggest the southern mountain alpine region in Sweden will be completely covered by forests at the end of the 21st century, making preservation of many rare and threatened species impossible. In the northern alpine regions, most vegetation types will persist but shift to higher elevations with reduced areal extent, endangering vulnerable species. Simulated potential for reindeer grazing indicates latitudinal differences, with higher potential in the south in the current climate. In the future these differences will diminish, as the potentials will increase in the north, especially for the summer grazing grounds. These combined results suggest significant shifts in vegetation composition over the present century for this scenario, with large implications for nature conservation, reindeer husbandry and forestry.

## 1 Introduction

High-latitude regions harbour vast areas of relatively intact ecosystems, holding species of great ecological and societal significance (Dobrowski et al., 2021). These northern ecosystems are predicted to be more vulnerable to climate change than most other terrestrial biomes (Hickler et al., 2012; IPCC, 2014). For each degree of global average temperature increase, the observed increase in Fennoscandia has been estimated to be 2-3 degrees (Rantanen et al., 2022), and this relationship persists in future predictions (Ono et al., 2022). This temperature increase has strongly affected northern ecosystems, resulting in changing vegetation patterns in the Arctic (Elmendorf et al., 2012; Pearson et al., 2013), an overall taller plant community (Bjorkman et al., 2018) and increases in biomass (Hudson and Henry, 2009). The occurrence and distribution of shrubs has also been observed to increase, both in high latitude and high-altitude regions, as a result of the warmer climate (Elmendorf et al., 2012; Myers-Smith et al., 2011; Sturm et al., 2001). The distance species have to migrate to keep up with climate change is, however, shorter in alpine and Oroarctic regions than in flat boreal and Arctic landscapes (Feeley et al., 2011). As the boreal forest covers a wide area, its species composition and ability to provide ecosystem services could undergo large shifts, e.g. as a response to different disturbance patterns and hydrology changes (Venäläinen et al., 2020), even if its geographical extent is not changed. Consequences of future shifts in areal extent of vegetation zones, which may not be proportional to their current distributions, include reduced space of many habitats (Pauli and Halloy, 2019) and increased pressure on many species (Kuuluvainen and Gauthier, 2018).

In Fennoscandia, the herding of semi-domesticated reindeer is important in shaping this landscape, a practice which utilises the land from the coastal areas and the boreal forest in winter up to the tundra in summer (Käyhkö and Horstkotte, 2017). Reindeer grazing directly affects the vegetation composition and diversity, both in the mountains (Olofsson et al., 2001; Sundqvist et al., 2019; Vowles et al., 2017) and forested regions (Kumpula et al., 2014). In summer, reindeer have a mixed diet of shrub leaves, forbs, herbs, sedges, grass, and fungal fruit bodies, and reindeer forage has been shown to reduce deciduous shrub expansion (e.g. Olofsson et al., 2001; Olofsson et al., 2009; Sundqvist et al., 2019; Vowles et al., 2017). In winter, reindeer mainly eat ground- and tree-lichens, which decreases ground-lichen cover (Kumpula et al., 2014). However, reindeer husbandry is currently experiencing increased pressure from human activities, such as forestry practices and tourism (Fohringer et al., 2021; Kumpula et al., 2014; Sandström et al., 2016), affecting 85% of the herding area (Stoessel et al., 2022). In addition, there are implications resulting from climate change, such as difficult snow conditions making winter forage hard (Rasmus et al., 2022; Rosqvist et al., 2021) and hot dry summers increasing heat stress (Käyhkö and Horstkotte, 2017). Climate change is increasing the pressure on both ecosystems and societies in these areas, a pressure that will increase in coming decades (Constable et al., 2022). Reindeer grazing and browsing can to some degree hold back the rate of tree-line advancement and tundra shrubification (Stark et al., 2023), but will not completely stop it. A future less open landscape will have a large impact on how reindeer graze and dwell in the landscape and on the Sami culture (Stark et al., 2023), but how that might affect the size of a sustainable reindeer population is hard to predict.

Projections of future impacts of climate change in high latitude ecosystems can be made upon the implementation of understanding arising from empirical studies (e.g. Bjorkman et al., 2020; Myers-Smith et al., 2011) and remote

sensing (e.g. Callaghan et al., 2022), into models such as dynamical vegetation models (DVMs) using climate model data as input. The typical cell size of a regional climate model (on the order of 10-50 km) often contains land surface types ranging from forest to bare rock or glaciers in mountainous areas. This information does not capture all local variation, especially in areas of complex terrain where altitudinal differences can be strongly underestimated (Lind et al., 2020). Also, while representing most meteorological processes some are only crudely implemented at such relatively coarse resolution in modelling studies (Lind et al., 2020). In recent years, one of these DVMs, LPJ-GUESS, has been adapted to the boreal and Arctic regions (Miller and Smith, 2012; Wolf et al., 2008; Yu et al., 2017), and used with very highly resolved climate data (e.g. $50 \times 50$ m) at a local scale in sub-arctic Scandinavia (Gustafson et al., 2021; Tang et al., 2015). So far, however, no high-resolution (<10 km) study of environmental change and its impact on vegetation covering the entire Fennoscandian boreal and Oroarctic region has been made.

Recently the first ever climate model projections at 3 km grid spacing were completed for the entire Fennoscandian region (Lind et al., 2020; Lind et al., 2022). Results from such km-scale simulations (1-4 km compared to previous coarser resolution >10 km), offer an unprecedented insight into weather and climate processes at high resolution, which is particularly important in complex terrain. In addition, it is important to understand how this improved weather and climate insights might affect vegetation dynamics and for that a DVM is needed. Thus, we here use these unique km-scale climate model projections for the high-emission RCP8.5 scenario (Lind et al., 2022) and the state-of-the-art DVM; LPJ-GUESS (Lindeskog et al., 2021; Smith et al., 2001; Smith et al., 2014), including a new module of reindeer grazing, to investigate the vegetation response to climate and environmental change in the Fennoscandian boreal and mountain regions used for reindeer herding. The results are validated against satellite products and field data gathered in the study region. Furthermore, we use consistent, high-resolution climate and nitrogen deposition scenarios to evaluate potential future vegetation changes in the region, with a special emphasis on reindeer food supply and vegetation trends in "hotspot" areas with high biodiversity and conservation values. As the only available climate projection at this resolution is a high-emission scenario, the simulated state at the end of the century will provide a message to society of what to expect and plan for if emissions continue to increase. We hypothesize that this state will show extensive changes that will present challenges for forestry, reindeer herding, tourism and nature conservation, as the temperature increase is high in such a scenario

## 2 Material and methods

### 2.1 Study area

The study was restricted to the Fennoscandian mountain range and the adjacent boreal areas used for reindeer herding (Figure 1), with a focus on ecosystems in Sweden. This region is located between 58 and 71 °N, spanning altitudes from sea level to 2469 m a.s.l. (Galdhöpiggen, Norway), and is characterised by continental to sub-oceanic climate (Oksanen and Virtanen, 1995). Boreal forest dominates from the coast towards the mountains up to latitude 68-69 °N. Above the boreal forest there is a zone of mountain birch forest which normally has a vertical distribution of ca 200 m. The tree-line, formed by mountain birch, is in Sweden at an altitude of more than 1100 m in the south and decreases with latitude to 600 m in the north (Kullman, 2016). Above the tree line follows

tundra with decreasing levels of vegetation height and coverage (from shrub- to barren tundra) and finally bare

rocks and snowfields. For a more detailed assessment of simulated changes, six "hotspot" areas (90 × 90 km) in

the larger domain were selected to represent different vegetation zones with a high species richness and large

conservation values, from the boreal forest to the high alpine tundra, and covering the entire Swedish mountain

range (Table 1, Figure 1, Figure S1):

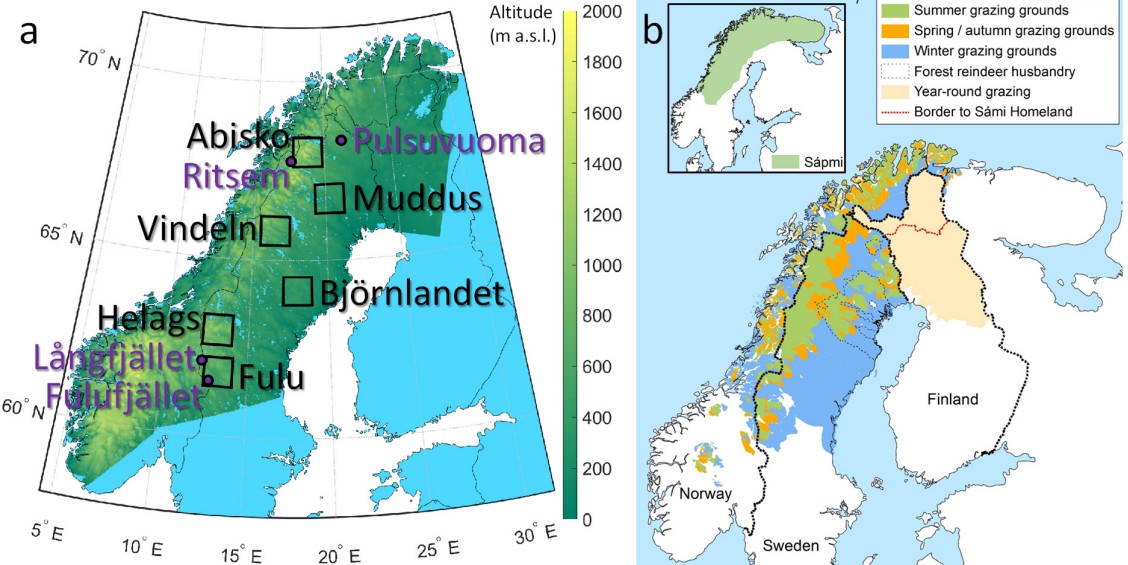

**Figure 1.** a) The study area (shown as altitude), the six focus "hotspot" areas (squares and black text, see Figure S1 for detailed

maps) and the four grazing exclosure sites (purple dots and text). b) Map of grazing areas used for the semi-domesticated

reindeer during different seasons in Norway, Sweden and Finland (from Käyhkö and Horstkotte (2017), used with permission).

**Table 1.** Description of the six "hotspots". Maps of the areas are shown in Figure S1.

| Name | Coordinates | Type | Protected area | Description |
|---|---|---|---|---|
| Abisko | 68° 01' N, 18° 46' E | Mountain | Abisko National Park, established 1909 | Including the highest mountains (2097 m) in Sweden |
| Vindeln | 65° 49' N, 16° 29' E | Mountain | Vindelfjällen Nature Reserve, established 1974 | Including mountains reaching 1768 m |
| Helags | 62° 58' N, 13° 06' E | Mountain | Vålådalen Nature Reserve, established 1988 | Including the mountain Helags (1797 m) |
| Fulu | 61° 47' N, 13° 17' E | Mountain | Fulufjället National Park, established 1973 | At the southernmost part of the Scandes mountains (1196 m, Sömlinghågna) in Sweden |
| Muddus | 66° 43' N, 20° 17' E | Forest | Muddus National Park, established 1942 | Mostly boreal forest with extensive wetlands |
| Björnlandet | 64° 07' N, 18° 01' E | Forest | Björnlandet National Park, established 1991 | Boreal forest with some wetlands |

## 2.2 Dynamical vegetation and ecosystem model

The dynamical vegetation and ecosystem model LPJ-GUESS (v4.1, Nord et al., 2021) was used to simulate vegetation change in the region. Detailed descriptions of the development stages involves; first version (Smith et al., 2001), arctic development (Miller and Smith, 2012; Wolf et al., 2008), N-cycle (Smith et al., 2014), landcover (Lindeskog et al., 2021) and further arctic implementations (Gustafson et al., 2021). The model simulates the development of cohorts, belonging to different plant functional types (PFTs), when competing for light, nitrogen and water in replicate patches (here set to 15 patches per simulated climate gridcell). Each patch represents an area of ca 1000 m$^2$. The model includes detailed process descriptions related to the cycling of water (e.g. transpiration, evaporation, and snow and soil water dynamics), carbon (e.g. photosynthesis, respiration, fire, and allocation of biomass), and nitrogen (e.g. nitrification and restriction of photosynthesis), and is driven by temperature, radiation, relative humidity, wind speed, $CO_2$ concentration, and nitrogen deposition data. Except those that take place in the soil, the processes are calculated at the cohort level. The PFTs are described by parameters related to growth form (tree, shrub or herbaceous), allocation, allometry, phenology, life history, shade tolerance and response to environmental and bioclimatic conditions. Patch destroying disturbances representing e.g. devastating pests or wind storms, occur randomly in each patch (the default v4.1 return time of 150 years was used in the presented simulations). Separately, fire disturbance simulated with the BLAZE module (Molinari et al., 2021) was applied. A simulation starts after a spin-up period (set to 600 years) over which a detrended dataset comprising the first 30 years of historical climate data are repeated to get the vegetation in balance with the climate.

## 2.2.1 Plant functional types

In the present study, an expanded set of PFTs were used, which includes high-latitude PFTs such as shrubs, with separate sets for mineral and wetland soils (Table 2). In each gridcell, the simulations on mineral and wetland soils are independent of each other.

For fractions of land classified as peatland, we use a version of the model with peatland integration (Wania et al., 2009a, b), which include a wetland hydrology module and wetland PFTs (Miller and Smith, 2012; Wolf et al., 2008; Zhang et al., 2013). The fractions of mineral soil and wetland were prescribed and constant over the simulation period based on the PEATMAP product at a 0.125° (14 km in S-N direction and 5-7 km in W-E direction depending on latitude within the assessed area) resolution (Xu et al., 2018). Weighted averages of model results were calculated based on these fractions. For the shade-intolerant broadleaved summergreen tree (IBS) PFT some parameters were changed according to Gustafson et al. (2021) in an application for Abisko, Sweden. Their revision was made to reflect the fact that this global PFT in Fennoscandia mainly represents mountain birch (*Betula pubescens* ssp. *tortuosa*). Details of the IBS parameterization are found in S2.

**Table 2.** Plant functional types in the LPJ-Guess simulations. The last six PFTs were used for the wetland simulation and the
rest for mineral soils.

| PFT | Long name | Typical represented species |
|---|---|---|
| BNE | Boreal needle-leaved evergreen tree, shade tolerant | *Picea abies* |
| BINE | Boreal needle-leaved evergreen tree, shade intolerant | *Pinus sylvestris* |
| IBS | Shade-intolerant broadleaved summergreen tree | *Betula pubescens* ssp. *tortuosa* |
| TeBS | Shade-tolerant temperate broadleaved summergreen tree | *Fagus*, *Quercus*, *Fraxinus* spp |
| C3G | Cool (C3) grass | *Poaceae* |
| HSE | Tall shrub (up to 2m), evergreen | *Juniperus communis* |
| HSS | Tall shrub (up to 2m), summergreen | *Alnus incana*, *Salix* spp. e.g. *S. phylifolia* and *S. myrsinifolia*, *Betula nana* |
| LSE | Low shrub (up to 0.5m), evergreen | *Vaccinium vitis-idaea*, *Empetrum* spp. |
| LSS | Low shrub (up to 0.5m), summergreen | *Vaccinium myrtillus*, small *Salix* spp. e.g. *S. arbuscula* |
| GRT | Graminoid and forb tundra | Grass, sedge and forb tundra species |
| EPDS | Evergreen prostrate (up to 0.2m) dwarf shrubs | *Vaccinium oxycoccos*, *Cassiope* spp., *Dryas octopetala*, *Saxifraga* spp. |
| SPDS | Summergreen prostrate (up to 0.2m) dwarf shrubs | Dwarf *Salix* spp. e.g. *S. herbacea*, *Arcto-staphylos alpinus* |
| CLM | Cushion forb, lichen and moss tundra | *Saxifragaceae*, *Caryo-phyllaceae*, *Draba* spp., lichens, mosses |
| pLSE | Peatland low shrub, evergreen | *Vaccinium vitis-idaea*, *Cassiope* spp. |
| pLSS | Peatland low shrub, summergreen | *Vaccinium myrtillus*, *V. uliginosum*, *Salix hastata*, *S. glauca* |
| pCLM | Peatland cushion forb, lichen and moss tundra | *Saxifragaceae*, *Caryophyllaceae*, *Papaver* spp., *Draba* spp., lichens, mosses |
| WetGRS | Cool, flood-tolerant (C3) grass | *Carex* spp., *Eriophorum* spp., *Juncus* spp., *Typha* spp. |
| pmoss | Peatland moss | *Spagnum* e.g. *S. fuscum* |
| C3G_wet | Peatland cool (C3) grass | *Poaceae* |


### 2.2.2 Reindeer grazing, browsing and trampling

To simulate the effect of reindeer grazing, browsing and trampling, a new module was added to the model. Graz-
ing/browsing was simulated by removing a fraction of leaf biomass. The model only includes stem and leaf as
above-ground compartments. We note however that when browsing reindeer also consume tops, twigs and branch
biomass, which mean that we may underestimate the effect on vegetation, but to keep the model uncomplicated
this simplification was applied. Trampling was simulated by killing a fraction of the individuals in a cohort, or, in
the case of herbaceous PFTs, a fraction of total biomass. The grazing/browsing and trampling level was based on
a constant intensity of herbivory. For a specific PFT, the grazing/browsing was determined by a preference value
obtained from extensive observations of the feeding preferences of semi-domesticated reindeer in Canada
(Denryter et al., 2017) and if the cohort's canopy height was within reach of reindeer. The sensitivity to trampling
was based on the vegetation response in an artificial trampling experiment (Egelkraut et al., 2020). All the con-
sumed carbon in the leaves was treated as harvested but only a fraction of the leaf nitrogen. The other fraction of
the consumed N was added to the cohort's leaf N pool, which reflects the assumption that N leaving the herbivore
as urine is rapidly taken up by the plants (Barthelemy et al., 2018). A detailed description of the module and its
parameter values is given in S3. From this module, the resulting output of consumed C biomass was used as an
indicator of potential reindeer food consumption. In the presented simulations, the simulated grazing, browsing
and trampling in a patch was set to have a return time of 3 years (see S3 for motivation), a grazing intensity of 0.1
(fraction $yr^{-1}$), a max height of 2.5 m and that 35% of browsed nitrogen (Ferraro et al., 2022; Mcewan and
Whitehead, 1970) was removed to the harvest pool.

**2.3 Model input data: climate**
The regional climate modelling system HCLIM38 (Belušić et al., 2020) was used for downscaling the RCP8.5
scenario simulation from the global climate model EC-Earth (Hazeleger et al., 2010; Hazeleger et al., 2012). The
climate scenario was first downscaled to 12 km with HCLIM38-ALADIN (Belušić et al., 2020) for the period
1985-2100 and then further to 3 km with HCLIM38-AROME (Belušić et al., 2020) for the periods 1985-2005,
2040-2060 and 2080-2100 (Lind et al., 2020; Lind et al., 2022). For computational reasons we wanted to restrict
the size of the complete 1985-2100 NetCDF climate files to less than 32 GB per climate variable. Therefore, a cut
of the original data was made at the south, north and west Norway-mainland limits as well as an eastern border
through the north of Finland (Figure 1).

The years 1985 in the ALADIN 12 km data and 1985, 2040 and 2080 in the AROME 3 km data were spin-up
years. To test the robustness of the results, all climate variables used by the vegetation model were also compiled
without using the HCLIM spin-up years and tested on a sub-set of 200 random gridcells. As there were no signif-
icant differences in the results we present results based on climate data including the HCLIM spin-up years.

For filling the periods when only ALADIN data were available, we first produced datasets such that the four
AROME gridcells coinciding with a certain ALADIN gridcell were filled with data from that ALADIN gridcell
(termed ALAatARO, 1985-2100). The periods with missing 3-km AROME data were filled with the ALAatARO
data using two methods:

For precipitation, global radiation, relative humidity, and wind speed, linear regressions through origin for the
overlapping periods between AROME data and the ALAatARO data were used. The relations were fitted sepa-
rately by month and, specifically, data from 1985-2005 and 2040-60 were used to establish the relationships for
the 2006-2039 period, and 2040-2060 and 2080-2100 for the 2061-2079 period. The relationships were then used
to get 3-km data for the missing periods from ALAatARO data.

For daily, minimum and maximum temperatures a non-parametric empirical quantile mapping "QUANT" bias
correction method (e.g. Osuch et al., 2017) was applied by month using daily temperature data for 21-year periods.
Two reference periods were used with observed 1 × 1 km data from Nordic Gridded Climate Dataset (NGCD,
https://surfobs.climate.copernicus.eu/dataaccess/access_ngcd.php) that were aggregated to the AROME grid,
1985-2005 (used for AROME grid data) and 1998-2018 (used for ALAatARO grid data). In the quantile mapping,
intervals of 1% were applied and a smoothing was done using a running mean over 5 intervals. Modelled and
matching observed values were linearly interpolated between the intervals. For consistency, all calculations of
quantiles were done for 21-year periods, resulting in an overlapping period (1998-2005) for which the AROME
data were used. For the future, the difference between observed and scenario quantiles during the reference period
was added to the matching quantile of future 21-year periods. The future periods were 2040-2060 and 2080-2100
for the AROME data and 2019-2039 and 2060-2080 (2060 and 2080 not used) for the ALAatARO data.

The RCP8.5 scenario used was the first dataset produced at this high resolution for the entire region. It is a scenario
with strongly increasing emissions of greenhouse gases, but the projection up to the mid-century is similar to
lower emission scenarios (Meinshausen et al., 2011). In the resultant daily air temperature data, the climate-change
signal was a 1.0-2.3 K increase in mean annual temperature from the 1991-2020 to the 2031-2060 30-year periods,
and a 2.5-5.2 K increase from 1991-2020 to 2071-2100 (Figure S4a-b). For annual precipitation the relative change
was -2.3 – 23.1% to 2031-2060 and -0.9 – 50.1% to 2071-2100 (Figure S4c-d).

**2.4 Model input data: soil texture, atmospheric nitrogen deposition and $CO_2$**
Soil texture data (clay and sand fraction) at 3 km resolution were taken from SURFEX (Masson et al., 2013), the
land surface model of AROME, ensuring consistency with LPJ-GUESS. These data originate from FAO soil
texture data at 0.0833° (9 km in S-N direction and 3-5 km in W-E) resolution (https://data.apps.fao.org/map/cat-
alog/static/search?keyword=DSMW).

Nitrogen deposition at monthly temporal resolution was used as input to LPJ-GUESS. The input was based on
two model simulations (MATCH-BIODIV and MATCH-ECLAIRE) with the Multi-scale Atmospheric Transport
and Chemistry (MATCH, Andersson et al., 2015; Andersson et al., 2007; Robertson et al., 1999) model. MATCH-
BIODIV (Andersson et al., Manuscript; Eichler et al., 2023) was forced by the climate simulation ALADIN at 12
km, and anthropogenic air pollution emissions from ECLIPSE V6b (Höglund-Isaksson et al., 2020). This data set
(https://previous.iiasa.ac.at/web/home/research/researchPrograms/air/ECLIPSEv6b.html, accessed Feb 2020) has
a resolution of 12 km and covers the period 1987-2051. MATCH-ECLAIRE (Engardt et al., 2017) was constructed
at 50 km resolution for 1900-2050, based on current climate and varying anthropogenic air pollutant emissions
ECLIPSE V4a and Lamarque et al. (2010).

MATCH-ECLAIRE was used to obtain 12 km resolution nitrogen deposition fields for the time period 1900-
1986. This was done by establishing a linear relationship through zero for the overlapping period for each
MATCH-BIODIV and MATCH-ECLAIRE gridcell and subsequently applying it to downscale the 50 km data
for the 1900-1986 period. After 2051, the 0.5° (56 km in S-N and 18-29 km in W-E) resolution Lamarque et al.
(2011) dataset was used, which is standard for LPJ-GUESS. A sensitivity test for a selection of gridcells showed
very minor differences when comparing modelling results for simulations using the different resolution of N dep-
osition data (results not shown), from which we concluded that this simplification was justified.

The future trend in nitrogen deposition is similar for MATCH-BIODIV and MATCH-ECLAIRE, i.e. declining
until mid-century. The modelled total deposition in the Scandinavian Mountain area is dominated by oxidized
nitrogen, which exhibits a clear decline, while reduced nitrogen deposition levels off at around 2020 and after that
even increases slightly for MATCH-BIODIV. These data sets have been evaluated against reanalysis data con-
sisting of fused observations and modelled estimates (Andersson et al., Manuscript) for years where all datasets
were overlapping (1987-2013) for high-altitude areas of the Scandinavian Mountains. The comparison shows a
positive bias in the quantitative modelled total nitrogen deposition by 18% and 23% respectively for the 1987-
2013 period, with a stronger positive bias in oxidised nitrogen deposition and a partly balancing negative bias in
reduced nitrogen. The trend is similar between the modelled and the reanalysed datasets over the period.

Historical and RCP8.5 $CO_2$ concentration data were the same as used by EC-Earth and HCLIM38, reaching at-
mospheric concentrations of 540.5 ppm and 935.9 ppm in 2050 and 2100, respectively (IPCC, 2013).

**2.5 Model output validation and analysis**
The output data from LPJ-GUESS are given as yearly states or sums of fluxes averaged over the 15 simulated
patches. The data were further averaged over 10-year periods to reduce fluctuations arising from interannual
weather variability and random disturbances.
**2.5.1 Validation**
The simulated total leaf-area index (LAI) was compared to yearly maximum of the monthly SURFEX LAI product
(Masson et al., 2013), taken from ECOCLIMAP2.2 based on MODIS at 1 km resolution in 2000 (Faroux et al.,
2013). Further, modelled LAIs for the specific PFTs within a gridcell were used to determine vegetation class for
comparison to two remote-sensing based vegetation products: the land cover of northern Eurasia (GLCE) based
on SPOT 4 at 1 km resolution (Bartalev et al., 2003) and the CLC2018 Corine land-cover dataset (Corine) based
on Sentinel 2 at 100 m resolution (https://land.copernicus.eu/pan-european/corine-land-cover/clc2018). The con-
versions (described in detail in S5) were based on Bartalev et al. (2003) and Kosztra et al. (2019) for GLCE and
Corine, respectively. The satellite-based products were aggregated to the 3 km AROME gridcells, based on dom-
inant class, to enable a statistical analysis of the agreement by means of confusion matrixes (e.g. Congalton, 1991),
with the satellite products used as ground truth. Three measures of accuracy were calculated. Producer accuracy
(PA, probability that a value in a given class was classified correctly, i.e. correctly predicted gridcells of a class /
total number of ground truth gridcells in that class), and user accuracy (UA, probability that a value predicted to
be in a certain class really is that class, i.e. correctly predicted gridcells of a class / total number of gridcells
predicted to be in the class) were calculated for each vegetation class, as well as the overall accuracy (sum of all
correctly classified gridcells for all classes / total number of gridcells).

The model output was also evaluated against ground-based data for biomass (trees and shrubs) and vegetation
coverage (field-layer) using data collected in 2011-2012 from four long-term exclosure experiments at Pulsu-
vuoma, Ritsem, Långfjället and Fulufjället (for Ritsem only field-layer coverage), all established in 1995 (Figure
1a) (Eriksson et al., 2007; Vowles et al., 2017). At each site and vegetation type (birch forest or shrub heath) there
were three fenced exclosure plots and three ambient plots of dimension 25 × 25 m. For the gridcells corresponding
to these experiments, the model was run both with a) continuous grazing and b) grazing stopped after 1995. To
convert model-simulated total biomass C to dry mass, a factor of 2.0 was used (Thomas and Martin, 2012), and
to convert modelled total biomass to above ground biomass we assumed a factor of 0.85 based on earlier estimates
for Swedish birch forest (between 0.79 and 0.92, Johansson, 2007). For seedlings, Huttunen et al. (2013) reported
values of about 0.6 increasing to 0.7 with fertilization and elevated temperature, but seedlings in general have
lower above-ground fraction (Qi et al., 2019). The vegetation cover data of the shrub and field layer, visually
estimated at species level, were aggregated to the LPJ-GUESS PFTs and compared to simulated LAI for 2-3 close
gridcells with similar altitude. Though the comparison of fractional plant cover and LAI is not strictly direct, the
two measures are closely related (George et al., 2021).

**2.5.2 Analysis**
Due to the more detailed adapted vegetation zones for boreal and Arctic conditions in the GLCE classification,
compared to Corine, the future trends presented in the results below focus only on the GLCE data.

To assess the simulated vegetation diversity, the Shannon (1948) Diversity Index (*D*) was calculated for each of
the six "hotspots" (Fig. 1) letting the number of gridcells of different vegetation classes represent diversity:
$D = -\sum(p_i \times ln(p_i))$                                                                                     (1)
Where *p* is the fraction of the total numbers of classified gridcells in the "hotspot" (excluding prescribed water
and wetland cells) belonging to class *i*. Only one vegetation class present would give a *D* of 0 and ten classes with
the same *p* would give a value of 2.3.

The simulated potential reindeer consumption of leaf carbon was aggregated to reindeer herding communities in
Sweden for traditional seasonal grazing grounds (Figure 1b), with help of GIS data obtained from the Swedish
Sami Parliament ([www.sametinget.se](www.sametinget.se)).

**2.6 Biodiversity data**
To investigate the sensitivity of the "hotspot" sites to change, species observations of all available species groups
together with threatened and red listed species, were extracted for each hotspot area using a GeoJSON file in the
"The Analysis portal for biodiversity data" database (downloaded 29[th] of October 2021. [https://www.analysispor-](https://www.analysispor-)
[tal.se/](tal.se/)). Further, to identify species being classified as alpine, the database "artfakta" ([https://artfakta.se/rodlistan](https://artfakta.se/rodlistan))
was used with selection criterion "Landscape type" set to "Mountainous".

## 3 Results

### 3.1 Validation

#### 3.1.1 Validation of simulated vegetation against satellite-based products

Simulated LAI was substantially lower than in the satellite based SURFEX product but had a reasonable agreement (Figure 2a-b, Simulated_LAI = 0.78 × SURFEX_LAI – 1.99, $r^2$ = 0.59). A reason for the low simulated values compared to the reference could be that yearly maximum of the SURFEX data were used, which can cause an overestimation if there are errors in the monthly data. Another aspect is that LAI is defined as one sided for SURFEX and projected in LPJ-GUESS, which corresponds to a factor of about 1.35 for needle leafed canopies (Flower-Ellis and Olsson, 1993).

The simulations capture most of the broad patterns seen in the vegetation distribution from forest to non-vegetated areas when compared to the satellite-based products in 2000 and 2018 (Figure 2c-f). For the detailed classes of the GLCE map the overall accuracy is however only 32% of the gridcells (Table S6a) and for the somewhat wider classes of Corine 37% (Table S6b). Classifying in broader classes, the extent of forest agreed for 84% of gridcells simulated to be forest for both GLCE and Corine (user accuracy, UA) and for 90% and 94% of the satellite-based forest gridcells (producer accuracy, PA) for GLCE and Corine respectively. The most common class of the boreal forest, the needle leaved evergreen forest class, is more mixed with broad-leaved trees in the simulation and the distribution west of the mountains is overestimated compared to the satellite-based products. With the new parametrization of the IBS PFT (Table S2), the deciduous broad-leaved forest expands too much in the north on the east side of the mountain ranges. Many (30%) of the gridcells that have shrub tundra according to the satellite data were classified as shrub vegetation, resulting in poor UA for broad-leaf deciduous shrubs (0.5%) and needle-leaf evergreen shrubs (0.0%), and poor PA for shrub tundra (0.2%), in the GLCE comparison. The classes are distinguished based on if the LAI of trees and tall shrubs is more than 20% of the total LAI (Figure S5b). Similarly, for Corine the simulated transitional woodland-shrub class mainly coincides with gridcells classified as broad-leaved forest, moors and heathland, and sparsely vegetated by the satellite product (Figure S5a).

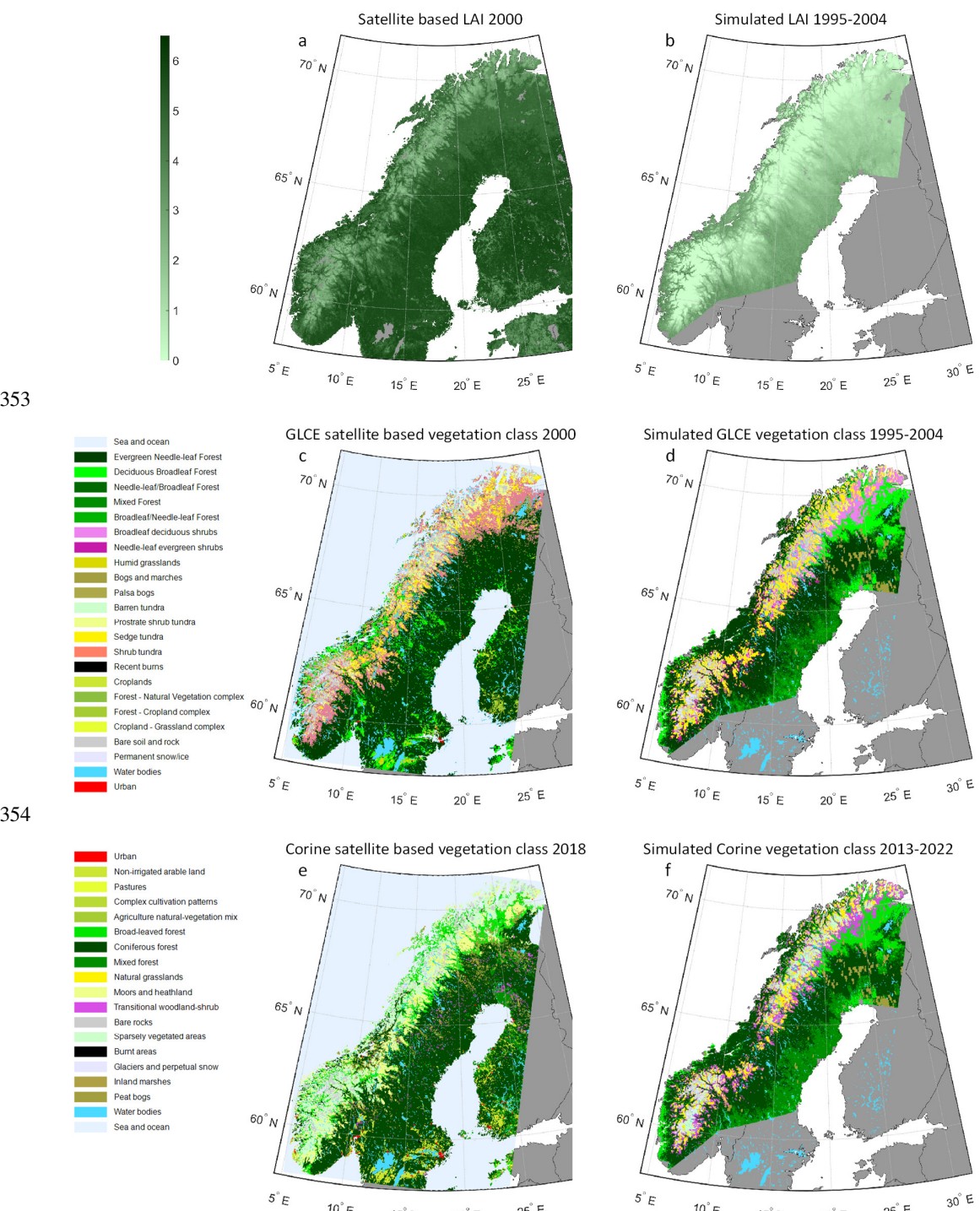




**Figure 2.** Satellite -based products of LAI (a), GLCE vegetation classes (c), and Corine vegetation classes (e) compared to simulated total LAI (b) and vegetation classes based on LAI averaged over 10-year periods for different PFTs according GLC Northern Eurasia (d) and Corine (f) (see Figure S5a-b).


Aggregating the tundra classes for GLCE gave a UA of 83% and PA of 36%, where the low PA is the result of
many gridcells classified as shrub tundra by the satellite data being simulated to be forest or shrub vegetation. The
"moors and heathland" class is the third largest in the Corine satellite data and was often classified as forest,
natural grassland or transitional woodland-shrub by the simulation (UA 18%, PA 4%). The LAI limit for the
definition of the bare rock and glacier classes from the simulation were the same for GLCE and Corine classifi-
cation and both classes were reduced in abundance from 1995-2004 to 2013-2022. The UA and PA for these
classes were in the range 11-57%.

**3.1.2 Validation of the effect of the new reindeer module against vegetation inventories and reindeer exclo-**
**sures**
Comparing model data and the *in-situ* estimated biomass for the northernmost Pulsuvuoma site showed that sim-
ulated tree and shrub biomass was underestimated by ca 35% but was within the inventoried uncertainty range of
the exclosure site (Figure 3). For the southern sites, biomass was overestimated by ca 50% at Långfjället and by
200% at Fulufjället. A reason for the substantial overestimation for Fulufjället is that it was dominated by needle-
leaf trees in the simulation. This was confirmed by test simulations; excluding pine and spruce PFTs (BINE and
BNE) reduced biomass with 14%, excluding also the birch PFT (IBS) reduced biomass with 78%.

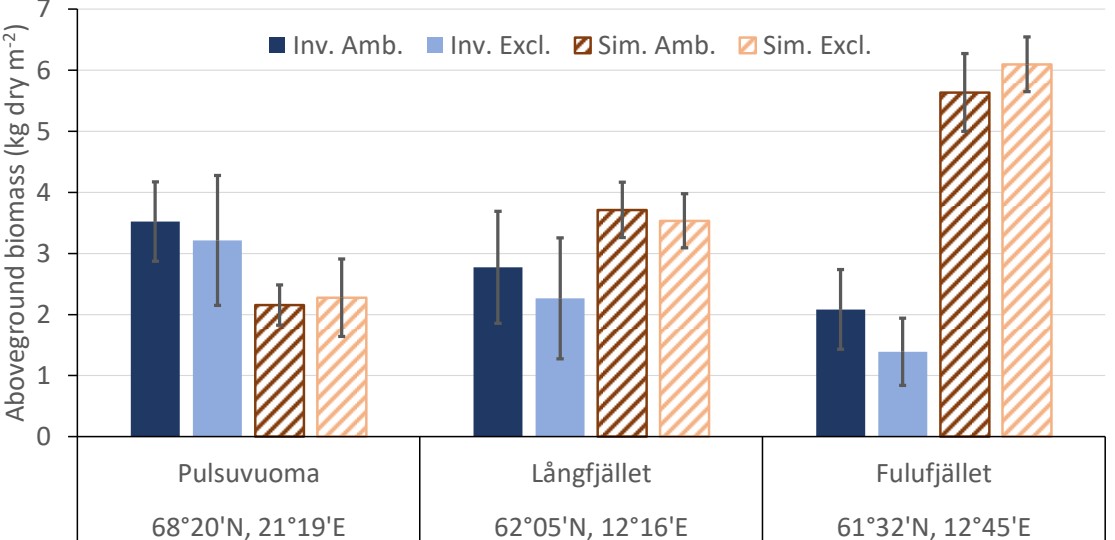


**Figure 3.** Simulated (Sim., mean over years 2009-2013, dashed bars) aboveground tree and shrub biomass compared to in-
ventoried data (Inv. 2011 or 2012, solid bars, no biomass data were available for the Ritsem exclosure site) from experiments
with ambient plots (Amb., dark bars) with reindeer access and plots with exclosure from 1995 (Excl., bright bars). Average
and standard deviation over 3 inventoried plots or the 2-3 closest simulated gridcells.

The *in-situ* observed coverage of the shrub and field layer from the four exclosure sites was dominated by low
evergreen shrubs, mainly *Calluna vulgaris*, *Empetrum nigrum* and *Vaccinium vitis-idaea*, except for the Ritsem
shrub heath, which was dominated by graminoids and herbs (Figure S6). The total simulated LAI of the shrub and
field layer was low for the two northern sites (0.08 to 0.26 compared to inventoried coverage of 59-75%) and was
dominated by graminoids and herbs in the Ritsem shrub heath and high summer-green shrubs below the
Pulsuvuoma birch forest. There was a trend that exclosure from reindeer grazing decreased the abundance of
graminoids and herbs in both observation and simulations. For the two southern sites, the inventoried coverage
and simulated LAI were similar except for Fulufjället, which had a simulated overstory of denser evergreen coni-
fers instead of birch. The trends after exclosure are less clear for the southern sites and the short shrub classes that
dominate in the inventories are almost totally absent in the simulations, which are dominated by high shrubs (up
to 2m tall), graminoids and herbs. It should be noted that Fulufjället is located outside the area used for reindeer
herding, though it is occasionally visited by reindeers from Norway and moose, and the modelling case is hypo-
thetical.

**3.2 Simulations and analysis of trends in vegetation 2000-2100**
**3.2.1 Trends in simulated vegetation classes over the whole simulated area**

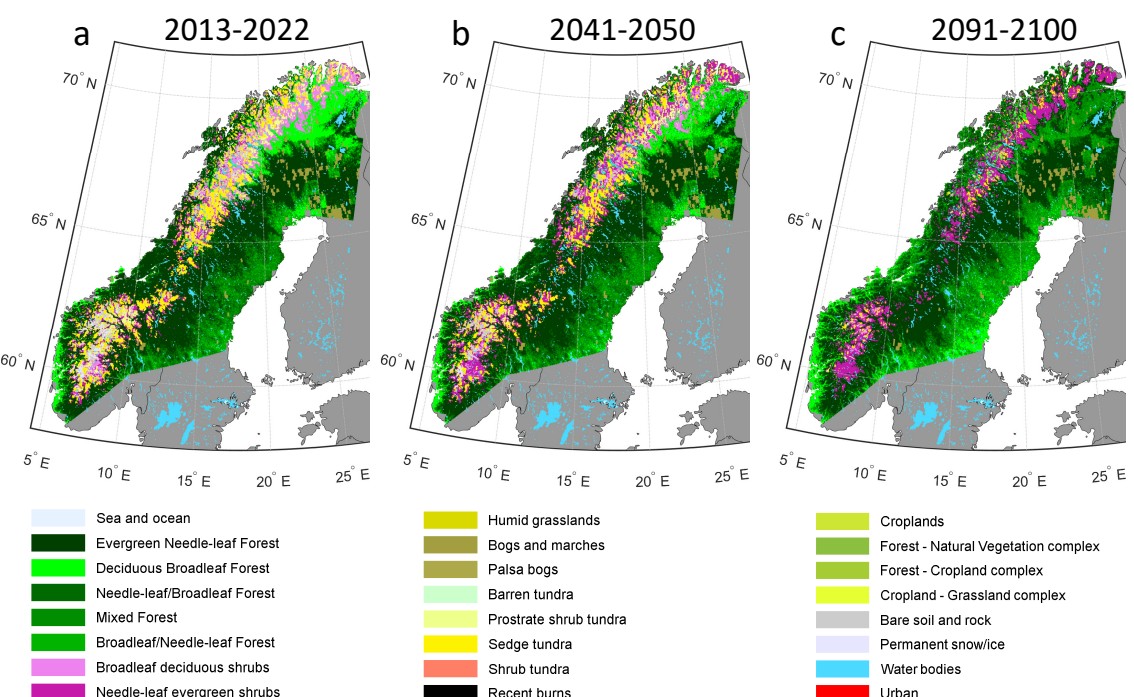


**Figure 4.** Simulated vegetation classes according to the GLC Northern Eurasia classification based on average LAI for differ-
ent PFTs over ten years, for three periods (a-c) in the RCP8.5 scenario.

In the RCP8.5 scenario a dramatic shift in simulated vegetation composition was found, especially after 2050
(Figure 4). By 2041-2050 the shrub vegetation classes are already seen to expand to higher elevations in the
mountains and the broad-leaved forests in the north start to be mixed with conifers. At the end of the century, the
simulated area coverage of open vegetation classes and the bare soil and rock class was found to be negligible.
For instance, the Fennoscandian Low Arctic tundra, which stretches like a wedge from the Kola Peninsula to
northernmost Swedish Lapland, in the lee of the mountain chain, would be completely lost by 2100 (Figure 4c).
Along the southern part of the Norwegian coast and the south-eastern part of the Swedish boreal forest, temperate
broadleaf trees (TeBS PFT) start to become dominant in the 2091-2100 period, shown by increasing areas of the
deciduous broadleaf class.

In the western part of the mountains at latitude 68° N, deciduous and mixed forests advance from a simulated
current maximum altitude of ca 500 m a.s.l. to more than 1200 m in 2090-2100 (Figure 5a-c). On the east side
there is no altitudinal advancement of the forest but a shift from deciduous broad-leaved trees to conifers. Shrub
vegetation classes, especially needle-leaved shrubs, become dominant at mid to high altitudes, for 66° N (Figure
5d-f) and 68° N at about 700 to 1200 m and for 61.5° N less distinctly at 1000 to 1700 m (Figure 5g-i). At latitude
61.5° N, the lower mountains east of the mountain range become almost completely covered by evergreen needle-
leaf forest. The changes seen in the 2041-2050 period are less distinct but the increase in needle-leaved shrubs
has started by then, and in the highest elevations a shift in gridcell classification from permanent snow/ice to bare
rock can be seen, indicating continued melt of glaciers and snowfields. As the classification is based on LAI, see
S5, it indicates that plants have the potential to grow there.

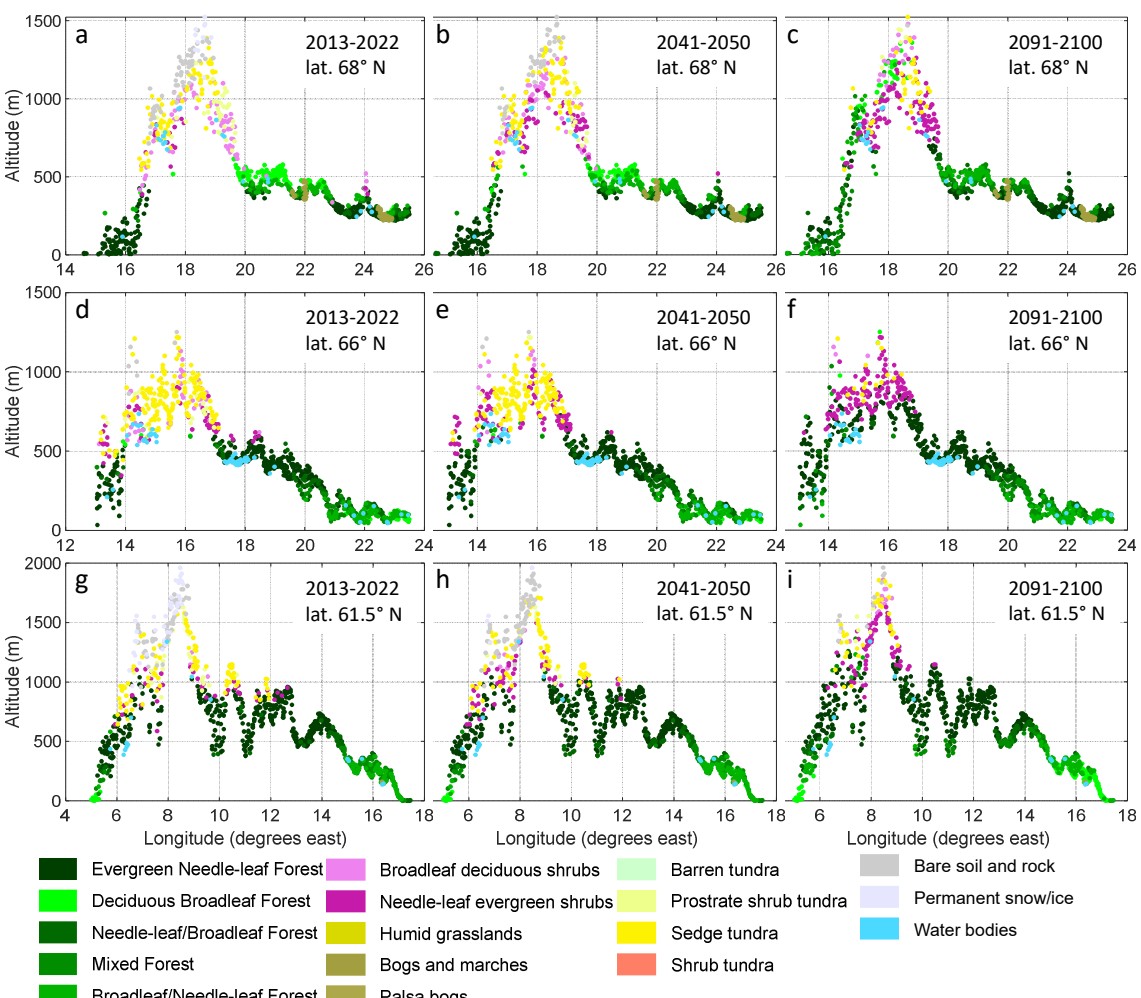


**Figure 5.** Profiles of simulated vegetation class according to GLCE for the 2013-2022 and RCP8.5 2041-2050 and 2091-2100
time periods, shown at the gridcells' longitude and altitude for three latitude bands at 68.0° N (a-c), 66.0° N (d-f) and 61.5° N.
The width of the bands (13.6-19.7 km) was set so that the area of each band was 9000 km² and contained about 1000 gridcells.

**3.2.2 Analysis of vegetation composition in "hotspots" with help of statistics of threatened species**

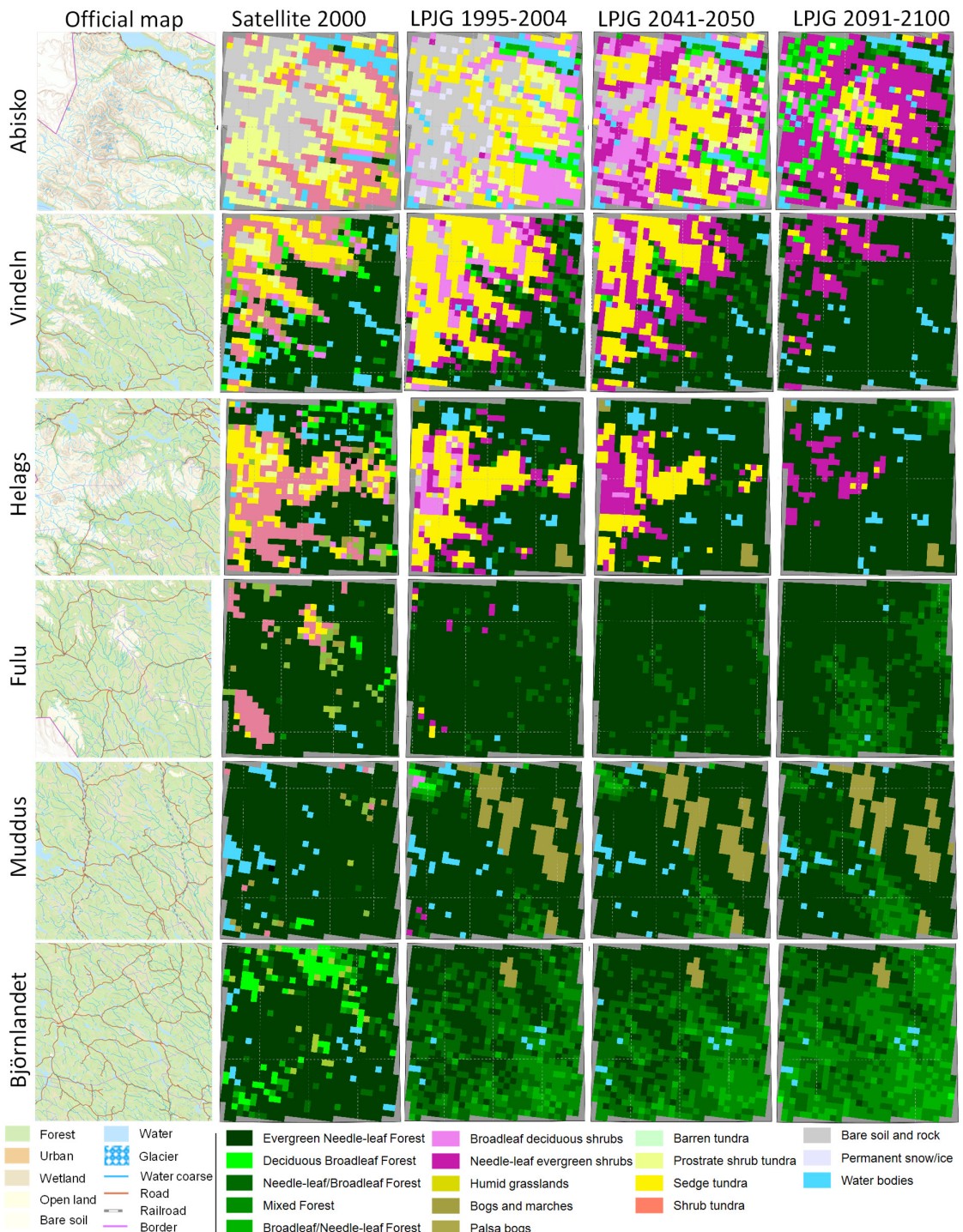

**Figure 6.** Satellite-based (GLCE, 2nd column) and simulated (column 3-5) vegetation composition in "hotspots" within four mountain areas (row 1-4) and two forest areas (row 5-6) (see Figure 1a for location), for 1995-2004 and for two future periods following RCP8.5. Each area is 90 × 90 km (30 × 30 gridcells). The first column shows the official vector-based map from Lantmäteriet (The overview map, open data license Creative Commons, (CC0), https://www.lantmateriet.se/en/, accessed 2021-09-09).

435

According to the GLCE satellite-based product, the shrub tundra class forms a large fraction of the vegetation next to the boreal needle-leaved forest (Figure 6, Table S7a-d). The official maps from Sweden (https://www.lantmateriet.se/en/maps-and-geographic-information/geodataprodukter/produktlista/oversik-tskartan/) show forest for parts of this area, e.g. in the valleys of the Abisko area.

440

For the Vindeln and Helags hotspots, the simulated distribution of forest was close to the satellite-based reference, but for the Fulu hotspot there are just a few gridcells simulated as vegetation other than boreal forest. For Björnlandet the mixture of broad-leaved trees in the forest is of similar magnitude but with a different pattern. The extent of sedge tundra was larger in the simulations than for the GLCE reference for the three northern mountain sites.

446

By 2041-2050 a significant shrubification occurs in Abisko, Vindeln and Helags (Figure 6), and forests start to establish at the edges of the current shrub and tundra vegetation, an advancement that accelerates until the 2091-2100 period.

450

In Abisko the needle-leaf shrub class reached a coverage of approx. 45% of the land area at the end of the century, expanding mainly over former broadleaf shrub, tundra and bare soil classes (Table S7a). In Vindeln and Helags the evergreen needle-leaf forest reaches approx. 80% coverage of the assessed area in 2091-2100 (Table S7b-c). In the boreal forest below the Fulu mountain and in the Muddus and Björnlandet areas, we see that the needle-leaf forest becomes more mixed with broad-leaved trees (Figure 6, Table S7d-f), which is also shown by higher Shannon Diversity Index ($D$, Table 3). It should be noted that the bog class, which is well represented in Muddus, is excluded from the calculation of $D$ as it is prescribed and not dynamic. Including this class would increase $D$ but we would not be able to correctly assess the change over time.

459

For the northernmost hotspot studied, Abisko, the bare soil and rock class will almost disappear in the RCP8.5 scenario, but most other classes will remain in similar proportions of the gridcells, though with a shift within the hotspot area (Table S7a). This is reflected in a minor increase in $D$ (Table 3) from 1.69 to 1.75 for this hotspot. Vindeln and Helags will see a clear decrease in diversity as needle leaved forest and shrubs will come to dominate (Table 3). For Fulu and the forests hotspots an increase in diversity is projected as the forests will be more mixed.

**Table 3.** The Shannon Diversity Index ($D$) calculated from the fractional cover of GLCE vegetation classes (see S7) of the "hotspots". Classes with non-dynamical vegetation like water and bogs were not included in the calculation.

|  | Abisko | Vindeln | Helags | Fulu | Muddus | Björnlandet |
|---|---|---|---|---|---|---|
| Satellite-based class 2000 | 1.44 | 1.38 | 1.42 | 0.50 | 0.14 | 0.80 |
| LPJ-GUESS simulation 1995-2004 | 1.69 | 1.72 | 1.25 | 0.32 | 0.50 | 1.19 |
| LPJ-GUESS simulation 2091-2100 | 1.75 | 0.66 | 0.52 | 0.83 | 0.65 | 1.29 |

468

Vindeln was the area with the lowest number of reported species, whereas Helags was the most diverse area with over 70% more species reported than for Vindeln (Table 4). The four other sites all had fairly equal numbers of

reported species, in the range of 5155-5647 species. However, all hotspots had a similar share of red listed species and threatened species, approximately 8-10% and 3-4%, respectively (Table 4).

Of all threatened species in Sweden (2764 species), only 5.2% (144 species) are classified as alpine and almost 2/3 of these threatened alpine species were found in Abisko, comprising more than half of all the threatened species in Abisko. For Vindeln and Helags, the number of threatened alpine species was just below 20%, whereas the southernmost mountain hotspot Fulu, together with the forest hotspots Muddus and Björnlandet, had less than 10% of their threatened species classified as alpine.

With respect to the species groups to which most of the threatened species belong, it can be noted that mosses contribute the largest number of species in Abisko (Table 4). Except Vindeln, where birds consist of the group with most threatened species, fungi represent the largest number of threatened species for the other four hotspot areas.

**Table 4.** Threatened species (VU=vulnerable, EN=endangered, CR=critically endangered) reported across species groups as well as total number of species and red-listed species reported for the six biodiversity hotspot areas.

| Species group | Abisko VU | EN | CR | Vindeln VU | EN | CR | Helags VU | EN | CR | Fulu VU | EN | CR | Muddus VU | EN | CR | Björnlandet VU | EN | CR |
|---|---|---|---|---|---|---|---|---|---|---|---|---|---|---|---|---|---|---|
| Birds | 20 | 11 | 3 | 24 | 14 | 3 | 28 | 13 | 3 | 23 | 14 | 3 | 25 | 14 | 4 | 20 | 10 | 2 |
| Fungi | 12 | 2 | | 27 | 2 | 1 | 57 | 7 | 1 | 52 | 9 | 4 | 63 | 10 | 2 | 39 | 5 | |
| Insects | 19 | 5 | | 22 | 1 | | 21 | 6 | | 19 | 3 | | 37 | 10 | | 23 | 11 | |
| Lichens | 8 | 2 | 1 | 16 | 4 | | 29 | 16 | 7 | 24 | 10 | 1 | 14 | 5 | | 14 | 4 | |
| Mosses | 40 | 9 | 1 | 6 | 1 | | 33 | 9 | | 16 | 6 | 1 | 5 | 3 | 1 | 5 | 2 | |
| Vascular plants | 18 | 7 | | 10 | 4 | | 17 | 11 | 1 | 26 | 12 | 3 | 8 | 5 | | 8 | 1 | 1 |
| Other groups | 1 | | | 1 | 1 | | 2 | 1 | | 1 | 1 | | 3 | 1 | | 2 | 1 | 1 |
| Threatened species (% of total) | 159 | 3.1% | | 137 | 3.4% | | 262 | 3.7% | | 229 | 4.4% | | 210 | 3.7% | | 149 | 2.8% | |
| of which are alpine species | 91 | 57% | | 25 | 18% | | 51 | 19% | | 20 | 9% | | 19 | 9% | | 9 | 6% | |
| Red listed species (% of total) | 423 | 8.2% | | 369 | 9.1% | | 651 | 9.3% | | 528 | 10% | | 547 | 9.7% | | 411 | 7.8% | |
| Total reported species | 5155 | | | 4058 | | | 7034 | | | 5205 | | | 5647 | | | 5250 | | |

**3.3 Simulations of reindeer presence**

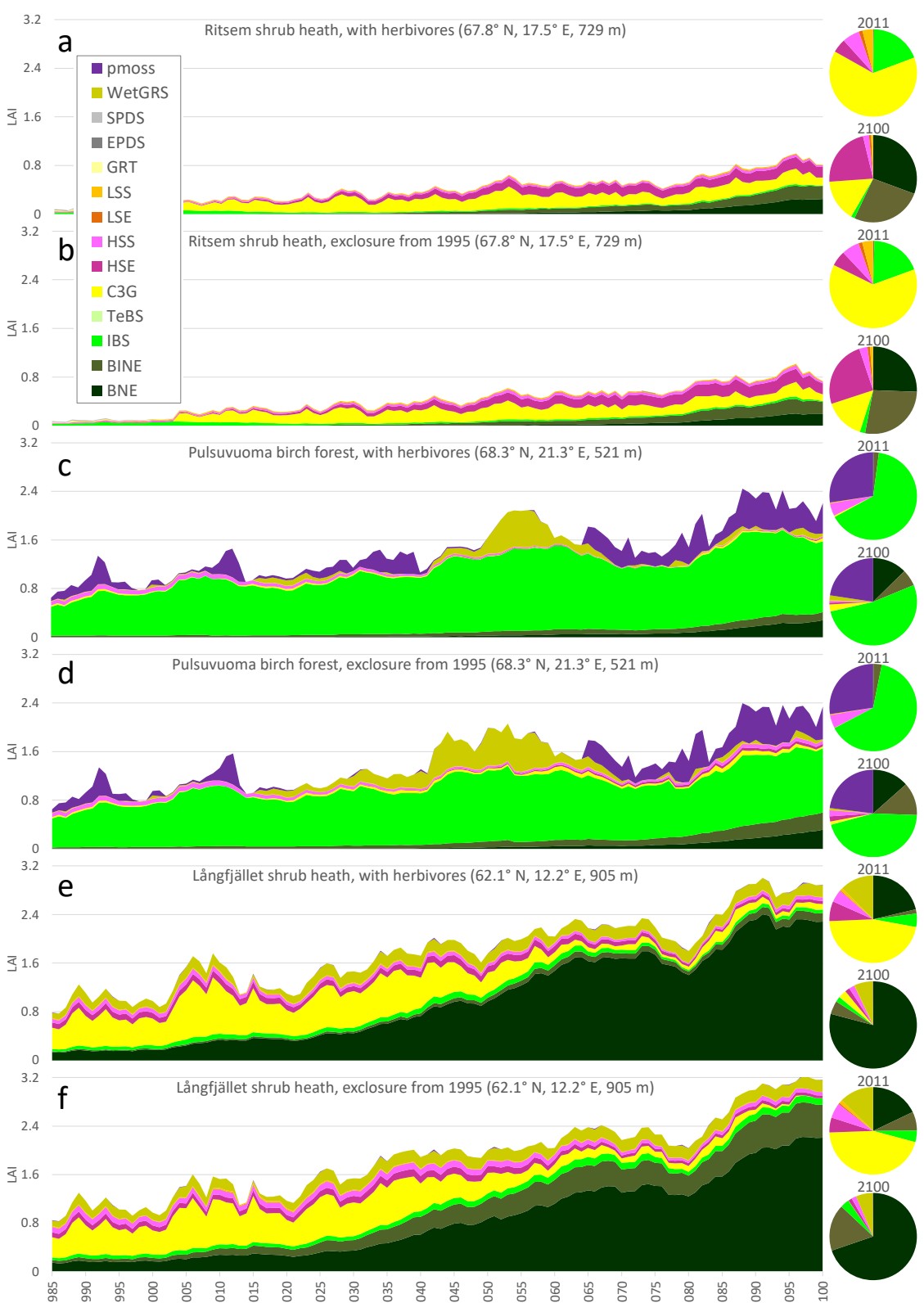

**Figure 7.** Simulated development of the vegetation composition (based on LAI for different PFTs, see Table 2 for description) at selected gridcells in the exclosure experiments 1985-2100, for RCP8.5.

### 3.3.1 Simulated effect on vegetation at reindeer exclosure sites 1985-2100

Three gridcells within the exclosure experiments with a wide range of conditions were selected to exemplify the simulated development of the vegetation composition until 2100 (Figure 7). Simulated LAI for the Ritsem shrub heath indicates a steep increase in year 2003, corresponding to an establishment of C3 grass, after which this PFT has a rather constant LAI over the simulation period (Figure 7a-b). Shrub vegetation (PFTs LSS, LSE (both low shrubs), HSS and HSE (tall shrubs)) increases gradually at Ritsem. At all sites, deciduous shrubs would have a higher fraction without simulated reindeer grazing and trampling but the difference is minor.

The mountain birch PFT (IBS) dominates simulations for the Pulsuvuoma birch forest over the simulation period (Figure 7c-d), but for the heath gridcells there is no period with a high fraction of mountain birch forests (Figure 7a-b, e-f). Instead PFTs that represent the needle-leaved coniferous forest (BNE and BINE) start to establish at the Ritsem and Pulsuvuoma gridcells around 2035 and these PFTs are already present in the simulations for Långfjället, and at the end of the simulation they are dominant at both shrub heath sites. The summer-green prostrate dwarf shrub PFT (SPDS) has a maximum fraction of ca 50% of LAI at Ritsem, though with a very sparse coverage, before C3 grass takes over, but apart from that, short shrubs (LSS and LSE), prostrate dwarf shrubs (SPDS and EPDS) and the graminoid and forb tundra (GRT) PFTs have only a minor presence in the simulations.

### 3.3.2 Trends in simulated potential reindeer leaf consumption 2000-2100

With a constant grazing pressure, simulated reindeer leaf consumption of a PFT depends on available leaf mass, accessibility of the leaves (height less than 2.5 m) and how appetizing it is (preference value – see Table S3). In the current climate the highest consumption was found east of the mountain range with an increasing gradient from north to south (Figure 8a).

In the boreal forest zone, the grazing level is quite evenly distributed, though there is a tendency for lower values in areas with a higher fraction of needle-leaf PFTs (Figure 4). The change by the 2041-2050 period is small, though some increased potential in the least vegetated areas can be seen (Figure 8b). By the end of the century there is a substantial increase in potential consumption in the higher altitude areas as well as in the inland boreal forest (Figure 8c). In the south and towards the east there is a trend towards reduced potential reindeer consumption in the forested cells.

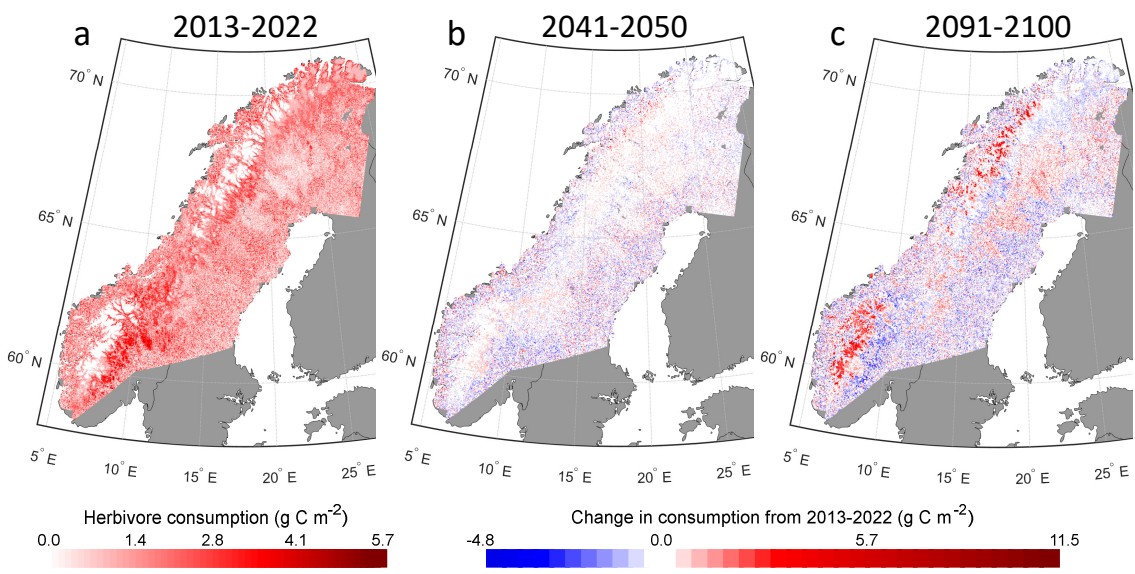

523

**Figure 8.** Simulated potential reindeer consumption (g C m$^{-2}$ yr$^{-1}$) 2013-2022 (a) and the change to 2041-2050 (b) and 2091-2100 (c) in RCP8.5.

526

The traditional spring and autumn grazing grounds of the Swedish reindeer-herding communities overlap to a high degree (Figure 1b) and both have a latitudinal dependency in potential reindeer consumption that is gradually reduced and eventually disappears by the end of the century (Figure 9a, c). For the summer grazing grounds there is a clear latitudinal dependency that is shifted in parallel (i.e. potential consumption increases uniformly) until the 2041-2050 period, but by the end of the century the latitudinal trend is gone or becomes negative, with higher potential consumption in the northern part of the study region (Figure 9b). For the winter grazing grounds, the latitudinal dependencies are weak and the southern communities have a trend of reduced potential grazing over time (Figure 9d). A more detailed compilation of the changes for the individual communities is given in S8.

535

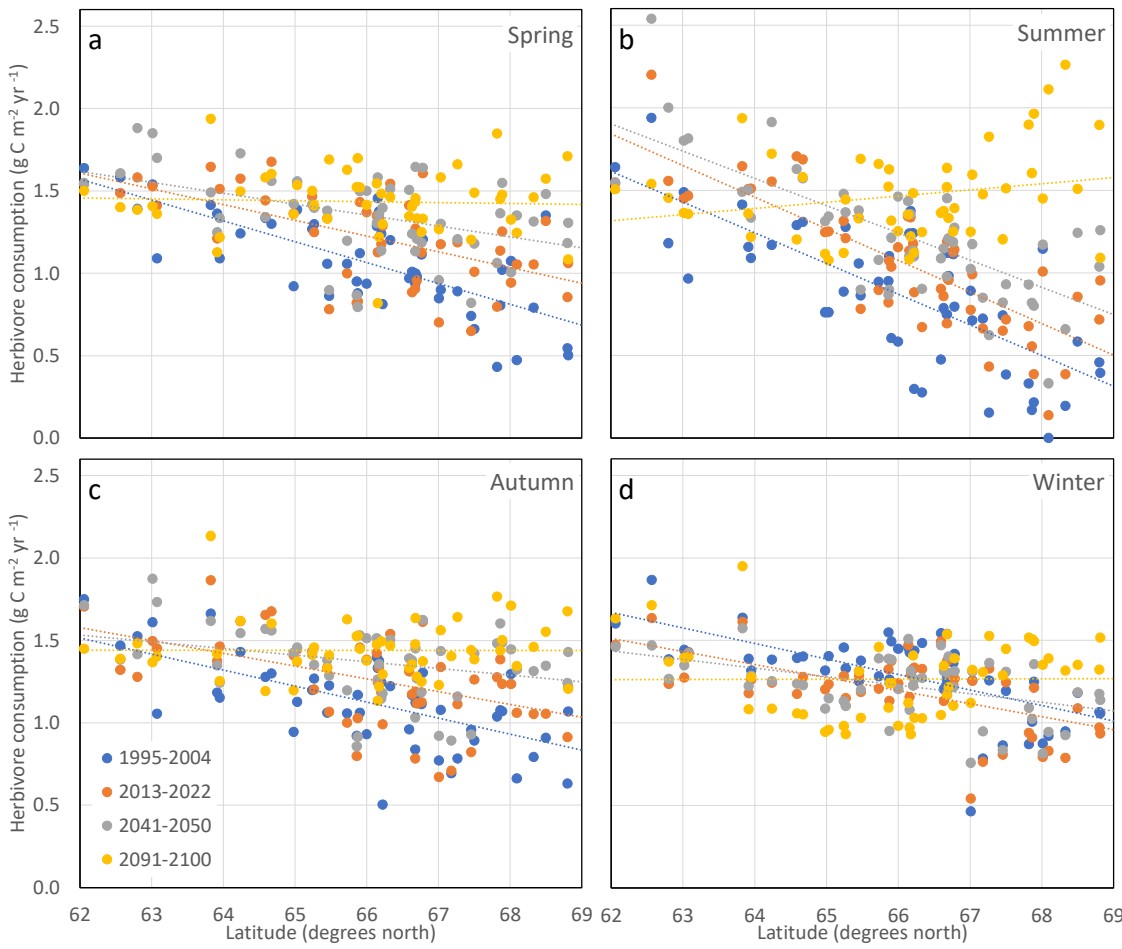

**Figure 9.** Simulated average potential reindeer consumption of leaf biomass in the 51 reindeer-herding communities in Sweden for the different seasonal grazing grounds (a-d) and four time periods (colour) in RCP8.5.

## 4 Discussion

The simulated changes in vegetation composition at the end of the century are extensive in our high-emission RCP8.5 scenario. For instance, we see a successive change in forest composition, from a cover of almost purely evergreen trees to a cover containing a larger fraction of broadleaved and mixed forest by the end of the century at the alpine Fulu and low elevation Muddus and Björnlandet hotspots. In Sweden, conifers are highly favoured by forestry for traditional and economic reasons, though pine forest regenerations are already encountering large problems (e.g. from moose grazing and diseases), which can further contribute to an increase of broadleaved forests in the future (Ara et al., 2022). Our results show that a profound vegetation change will occur at the southern alpine hotspots Vindeln, Helags, and Fulu, with the most dramatic changes projected for Helags and Vindeln. Here, a rapid tree growth and expansion is observed in this scenario, with only a few tundra-denoted grids remaining by 2091-2100. This change is also associated with a strong reduction in landscape diversity, as indicated by a decrease in the vegetation-class based Shannon diversity index. Today, the largest continuous Fennoscandian Low

Arctic tundra areas are found between the Swedish high mountains and the border between Finland and Sweden at latitudes around 68.5 °N and in the northern and western parts of the Finnmarksvidda plateau in the north of Norway. In a changed climate, the edges of the tundra area have probably become "Scandinavianized" (Vuorinen et al., 2017), i.e. the coverage of dwarf birches and lichens has decreased, while the *Ericaceae* species have increased. In the simulations, the tundra remaining in Helags and Vindeln will be dominated by needle-leaf evergreen shrubs with just a few scattered sedge tundra areas (e.g., wet tundra areas). These results are similar to the results from long-term warming experiments and monitoring plots in the northern Scandes, where most communities showed a "heathification" with time, both in the experimental warming and under ambient conditions subjected to the ongoing temperature increase (Scharn et al., 2021). At Fulu, the relatively extensive alpine tundra areas are situated just above the tree-line today, and here the tundra will be completely lost following RCP8.5. Thus, with a continued warming of up to 5 K to the end of the century in our model domain, which is not totally unlikely as the world is currently on track for a 2.9 K global warming (United-Nations-Environment-Programme, 2023) and given the Arctic warming amplification (Rantanen et al., 2022), the Fennoscandian vegetation is likely to undergo rapid shifts.

The simulated change in the extent of vegetation zones is driven by establishment of PFTs, but the richness of newly established vegetation depends on the migration of all associated types of organisms. The distance species need to spread to keep up with the shifts in climate is much shorter in mountainous than in flat regions, and since the ability to spread and inhabit new regions varies among species, a loss in species richness only occurs if new immigrants are stronger competitors than the intrinsic species (Pauli and Halloy, 2019). Though total reported species richness is largest in the Helags biodiversity hotspot area and lowest in Vindeln, both have equal fractions of alpine species. Although being the best available database, it should be kept in mind that the data obtained from the Analysis Portal relies on what has been reported by a large community of public and professional naturalists, which means that biases can exist e.g., depending on the specific biological interests of rapporteurs visiting the different areas. As they share the same trajectories, it seems likely that the homogenization of the vegetation composition in the Helags and Vindeln areas will lead to profound shifts in the conditions for many species, especially for the alpine species occurring here. In contrast to the southern alpine hotspots, our modelling results indicate that the northernmost hotspot area Abisko likely will retain large areas of alpine vegetation at higher elevations and its landscape diversity could even slightly increase. A substantial transformation of the vegetation cover is however also expected for Abisko. This includes shrubification, a process that has already been observed in this region (Hedenås et al., 2011; Rundqvist et al., 2011; Scharn et al., 2022; Scharn et al., 2021), and the broadleaved forest moving up well above 1000 m a.s.l. from the current level of about 600-800 m (Callaghan et al., 2013), a treeline advance that also been noted in regional high resolution simulations of Abisko (Gustafson et al., 2021). Abisko is the hotspot with the largest fraction of threatened alpine species in our study, and given the large elevation span in the region there are possibilities that some species may survive in microrefugia (Mee and Moore, 2014). Our results imply that a revision of the red-list and threatened species categories is urgent. This is because many of the alpine species in the hotspots areas that are not listed today will be threatened as warming continues (Schwager and Berg, 2019).

The simulated potential reindeer consumption shows a striking increase in the summer grazing ground north of
ca 65.5 °N. Although the simulated potential reindeer consumption is high, it is in the range of what can be
estimated from the current reindeer population in Sweden. Today, reindeer husbandry is practiced in about 50%
of the Swedish land area (i.e. 225 000 km$^2$, www.sametinget.se/rennaring_sverige) and the population is 225 000
– 280 000 animals in winter (www.sametinget.se/rennaring_sverige). With a consumption of 3-5 kg biomass per
reindeer and day (Yu et al., 2017), this equals an average total consumption over the area of about 0.8 g C m$^{-2}$ yr$^-$
$^1$, a number likely underestimated as the livestock is larger in summer before autumn slaughter. However, in our
simulations of potential reindeer winter consumption, the trends were weak both in latitude and time. Using a
constant herbivory intensity in the simulations means that the potential reindeer consumption shown represents a
hypothetical case in which we investigate how much would be consumed of the amount that is actually present if
the same number of reindeers and the same amount of food of the same quality is present in all gridcells. This
means that we have not considered mitigation and adaptation factors that may be of great importance such as
climate feedbacks on the population size and changes in what land areas the reindeer feed (Bråthen et al., 2017;
Speed et al., 2019). Reindeer grazing, browsing and trampling may also cause indirect feedback to the climate by
affecting the vegetation and ground properties, which in turn affects snow cover, albedo, carbon cycling and
biogenic volatile organic compound emissions (Brachmann et al., 2023; Holmgren et al., 2023). The representa-
tion of available reindeer food in the forested winter grazing grounds is challenging. In our simulations, the po-
tential reindeer consumption mainly consists of grasses that occur for a period after the random disturbances (with
an average 150-year interval), but grasses are not the preferred reindeer food during winter. Instead, reindeer eat
lichens in winter, which naturally can form dense layers under forests in the region. Current forest management,
creating a dense and uniform tree cover, disturbs the growth of lichens (Kumpula et al., 2014). Furthermore, our
weak trends during winter also depend on a delicate balance between a general increased productivity and higher
density of the tree canopies. This balance is also important for the implementation of ground lichen PFTs, since
there is a negative relationship between forest density and lichen abundance (Sandström et al., 2016). Thus, future
improvements to simulations considering reindeer grazing would need: a better representation of winter forage by
developing a new lichen PFT (e.g. Porada et al., 2016); an improved light-interception scheme; forest management
functionality and scenarios (e.g. Lindeskog et al., 2021); and a representation of restricted access to the field and
bottom layer vegetation during periods of difficult snow conditions. Though the simulated potential reindeer con-
sumption does not show dramatic shifts over the simulated period, reindeer herding will nevertheless experience
climate and weather related challenges in the future with e.g. concerns for hot and dry summers, more frequent
freeze-thaw cycles and rain-on snow events during winters, as well as expanding and denser forests (Käyhkö and
Horstkotte, 2017; Rosqvist et al., 2021). Thus, to be able to tackle and understand future challenges for reindeer
herding this not only suggests a need to include trophic interactions in models, but it also suggests that it is crucial
to evaluate the impact of extreme events on other important aspects of the environment for reindeer herding than
vegetation state alone.

We show the benefit of using high-resolution climate data to drive our DVM, enabling the simulation of a diverse
landscape, exemplified by our hotspot analysis (which would have less than 4 gridcells at a typical RCM resolution
of 50×50 km). Climate representation has also improved. In particular, the simulated precipitation patterns in
coastal and mountain areas as well as the ratio between snow and rainfall at high altitude show significantly better

agreement with observations at higher resolutions (Lind et al., 2020). Thus, highly resolved climate data in combination with a state-of-the-art dynamical vegetation model clearly contributes to a better understanding of climate-vegetation interactions in mountainous regions.

Using the detailed classification from GLCE, the accuracy scores for the simulated vegetation classes compared to the satellite product are low. For such a large area and high resolution as in the present study, an evaluation against satellite products is the only alternative with a complete coverage, but the satellite classes cannot be considered a real "ground truth". An example of possible misclassification of the GLCE data is clear from the fact that the mountain-birch forest in some of the valleys is classified as shrub vegetation, most clearly seen for Abisko and Vindeln when compared to the official vector-based maps. The shrub and tundra ecosystems have many subclasses and the model has some difficulty in reproducing the satellite-based pattern for these. Furthermore, the parameterization of the PFTs representing these systems is based on global or regional implementations driven by monthly climate data at coarse spatial scale (Wolf et al., 2008; Zhang et al., 2013), and it is not surprising that the results call for some model adjustment. The comparisons of results from the simulations with and without reindeer presence against exclosure site data do not show conclusive results. By only remove leaf biomass, we may have underestimated the effect of browsing, which particularly effects the height development of deciduous shrubs (Vowles et al., 2017). To study the effect of reindeer exclosures *in situ* also has some experimental constraints and uncertainty (Stark et al., 2023), e.g. as also other grazers like the hare or the lemming can have significant impact on the vegetation (Olofsson et al., 2012; Vowles et al., 2016). A further limitation of the vegetation simulations is that a soil layer always is present. The strong expansion of shrubs on former "bare soil & rock" and "permanent snow/ice" classes, e.g. as predicted for the Abisko area, is, therefore, probably overestimated, and instead parts of this area would become some type of tundra associated with shallow soils. Dispersal capacity and fire disturbance are also factors that may restrict vegetation expansion, as integration of those processes in an extrapolation of current trends in Alaska and western Canada reduced the predicted shrub expansion on non-shrub tundra from 39 to 25% by 2100 (Liu et al., 2022). In LPJ-GUESS, new PFTs can establish when climatic conditions are met, as we assume seeds are always present. Taking seed dispersal into account will in general slow down migration rates (Epstein et al., 2007; Zani et al., 2022). In mountainous terrain the distances will, however, be shorter and less dependent on dispersal capacity, e.g. reflected in abounded observations of tree seedlings more than 100 m above the treeline in Scandinavian mountains (Hofgaard et al., 2009). The disturbance return interval is an important and uncertain parameter that varies in time and space, which affects carbon stocks, the balance between shade tolerant and intolerant species as well as plant establishment (Pugh et al., 2019). Recent studies have suggested longer intervals (Pugh et al., 2019), but there are also studies that show fire return intervals of 50-90 years in boreal forest in Sweden (Dubber et al., 2017). There are also a lot of other activities that may prevent the establishment of new plants like soil processes, seed predation, plant browsing and mortality by smaller animals like rodents and hares, pests, pathogens, snow damage, moose, et cetera, which could be of potential importance. There is also a positive bias in the nitrogen deposition scenario (Andersson et al., 2023, manuscript) that could have further enhanced the simulated rate at which higher vegetation types expand (Gustafson et al., 2021). In the boreal forest region, the simulations have a higher fraction of broadleaf trees than the reference. A reason for this is that more than 90% of these forests are managed and needle-leaved trees are favoured in planting and thinning (Hannerz and Ekström, 2021) whereas the simulations represent natural, unmanaged vegetation where

broadleaf trees are common during the regeneration phase after disturbances in boreal forest (Angelstam and
Kuuluvainen, 2004). However, notwithstanding these limitations, our simulations clearly show that for Fen-
noscandia, the RCP8.5 pathway results in more prominent temperate features in the boreal forest, and that these
will expand northwards and to higher altitude resulting in a significant loss in tundra.

**5 Conclusions**

Our application of highly resolved climate data greatly improved both the representation of climate conditions
and the variation in simulated vegetation in mountainous landscapes. Climate and environmental change con-
sistent with the high-emission RCP8.5 scenario could cause dramatic shifts in the vegetation composition of the
Fennoscandian boreal and mountain regions, with consequences for reindeer herding, forestry and tourism sectors,
how we should practise conservation, and how we should manage our northern ecosystems. Indeed, these changes
have already started and been observed, but they will accelerate during the 21$^{st}$ century. Following a climate
trajectory in line with RCP8.5, the southern and lower elevation parts of the Fennoscandian mountain range that
today have tundra vegetation will be covered by forests in the coming century, while high-elevation regions will
undergo intense shrubification. In the northern tundra regions, most vegetation types will still be present at the
end of the century but shift in altitude and be compressed to smaller regions. This will threaten already vulnerable
species, especially those with slow dispersal rates and low competitive ability. In the southern part of the study
area a massive loss of alpine habitats and species is expected. The question is rather what new vegetation types
and species could occupy this area under continued climate change. There is, however, uncertainty at many levels
in this type of study: What emission scenario will the future hold and is it adequately interpreted by the global and
regional climate models? Is the vegetation's direct response to climate, $CO_2$ concentration, and nitrogen deposition
adequately described in the DVM? How will secondary effects of climate change alter disturbance patterns and
land use? Due to computational limitations in this high-resolution application, it has not been possible to quantify
these uncertainties (e.g. we only have one climate scenario), but it is clear from our results and those from previous
studies that all these aspects are important. The direction in which the results point is, however, clear in most
aspects. The rate of actual vegetation changes will also depend on factors such as forest management, reindeer
husbandry, other disturbances (such as fire) and the dispersal rate of different species. Our results indicate trends
towards increasing amounts of suitable reindeer forage, at least in northern Sweden, but other changes resulting
from climate change, such as the extent of open landscapes, heat stress and altered snow conditions are likely to
impact reindeer herding practises more than forage availability. The expected and potentially additive pressures
of environmental changes call for scenario-based research where the main drivers of the development, including
climate change, air pollution, land use and ecological processes, are considered in a consistent framework.

**Code availability**

The LPJ-GUESS code used and developed in this study is archived in the LPJ-GUESS Community Repository on Zenodo: https://zenodo.org/record/8262590 (Lagergren et al., 2023). More information about the model can be found at https://web.nateko.lu.se/lpj-guess (LPJ-GUESS developers, 2021).

**Data availability**

A selection of the MATCH-BIODIV dataset and the MATCH-ECLAIRE (Engardt et al., 2017) datasets are archived in Zenodo (MATCH-BIODIV: https://zenodo.org/record/7573171 and MATCH-ECLAIRE: https://zenodo.org/record/4501636#.ZBqvOXbMJaQ). The ALADIN and AROME HCLIM climate datasets (Lind et al., 2022), and the complete MATCH-BIODIV nitrogen deposition dataset (Andersson et al., 2023, manuscript) were generously shared with the authors but are not publicly accessible; the data can be accessed upon inquiry to the authors. The ECLIPSE V6b nitrogen deposition data are available from IIASA (https://previous.iiasa.ac.at/web/home/research/researchPrograms/air/ECLIPSEv6b.html) and the NGCD data used for bias correction of temperature can be accessed at the MET Norway Thredds Service (https://thredds.met.no/thredds/catalog/ngcd/catalog.html). The FAO soil texture data are available at the SURFEX site (https://www.umr-cnrm.fr/surfex/spip.php?article135). The Corine land-cover data (https://land.copernicus.eu/pan-european/corine-land-cover/clc2018) and the GLCE product for northern Eurasia (https://forobs.jrc.ec.europa.eu/products/glc2000/products.php) are freely available. The GIS data of reindeer herding communities were obtained after personal contact with Peter Benson from the Swedish Sami Parliament (www.sametinget.se) but are not freely available. Vegetation cover data (Vowles et al., 2017) can be accessed through Environment Climate Data Sweden (https://doi.org/10.5879/ECDS/2017-01-29.1/0). The biomass data from the exclosure sites are not available to the public but can be accessed by personal contact with the authors (R. Björk). Species observations for the hotspots are available at "The Analysis portal for biodiversity data" database (https://www.analysisportal.se/). Model simulation results with LPJ-GUESS for this manuscript are stored on DataGURU: https://dataguru.lu.se/app#BioDiv-S (Lagergren and Miller, 2023).

**Author contribution**

FL and PAM designed the study with contribution from RGB, CA, MPB, EK, HP and GR. FL carried out the vegetation model development, setup, runs and data analysis with support from PAM. RGB extracted and analysed the biodiversity data with help from MPB and HP. DB, PL and DL provided the climate scenario and associated soil and vegetation attributes. CA and TO provided high-resolution nitrogen deposition data. FL carried out bias correction and filling of continuous climate and nitrogen deposition data with advice from DB, EK and CA. RGB, MPB and GR contributed with expertise in reindeer husbandry and its interaction with the vegetation. FL prepared the manuscript with input from all co-authors.

**Short summary**

The Fennoscandian boreal and mountain regions harbour a wide range of ecosystems sensitive to climate change. A new, highly resolved high-emission climate scenario enabled modelling of the vegetation development in this region at high resolution for the 21st century. The results show dramatic south to north and low to high altitude shifts of vegetation zones, especially for the open tundra environments, that will have large implications for nature conservation, reindeer husbandry and forestry.

**Acknowledgement**

This work was supported by the BioDiv-Support project funded through the 2017-2018 Belmont Forum and Bi-odivERsA joint call for research proposals, under the BiodivScen ERA-Net COFUND programme, and with the funding organisations AKA (contract no 326328), ANR (ANR-18-EBI4-0007), BMBF (KFZ: 01LC1810A), FORMAS (contract no:s 2018-02434, 2018-02436, 2018-02437, 2018-02438) and MICINN (through APCIN: PCI2018-093149). The work is a contribution to the strategic research areas MERGE and BECC, and the profile area Nature-based Future Solutions hosted by Lund University. We thank Peter Benson at Sametinget for provid-ing data of the reindeer husbandry districts in Sweden and Mora Aronsson, Debora Arlt, and Johan Nilsson for advice regarding the extraction of data from the "The Analysis portal for biodiversity data".

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
