# Peer review of "Kilometre-scale simulations over Fennoscandia reveal a large"

_Biogeosciences, 2023_

## Author Comment (AC1)

- **RC1**: 'Comment on bg-2023-148', Anonymous Referee #1, 06 Nov 2023

**Common points:**

The paper aims to model the vegetation development until 2100 in the Fennoscandinavian to Oroarctic environment due to climate change in a high resolution, including the impact of reindeer grazing/trampling and the effects on biodiversity. The aims of the paper fit well to the scopes of the journal.

We thank you for careful reading and many useful comments that will improve the paper.

In my opinion the resolution is not high. High resolution should be 100 m to 10 m. The data presented here have a resolution of 3 km. Furthermore, the study area extends from the boreal forest to the tundra. Therefore, I would suggest adjusting the title.

What high resolution means depends on what field you are working in. In the context of regional climate modelling, as well as large-scale combined climate-vegetation modelling, 3 km represents a higher resolution than in most other studies. However, we agree that you have a point about the scale and will revise. We will also revise the title to emphasize that simulations were done over a large area. New title will be "Kilometre-scale simulations over Fennoscandia reveal a large loss of tundra due to climate change".

The second paragraph of the introduction should be moved to the description of the study area.

We will keep a smaller introduction of the region to introduce the next paragraph. But most of the text will be merged with the first paragraph of the M&M in line with the reviewer comment.

In the last paragraph of the introduction (Line 93 to 100) it could improve the manuscript (structure) if the task and the objectives are clearly stated. The objectives should then be always used in the same order in the other sections of the paper (methods, results, and discussion).

As also the other reviewers have pointed out that task, objective and hypothesis should be clearly stated, we will revise the last section of the introduction. We will add this sentence after "which is particularly important in complex terrain.": "It is important to understand how climate change might affect the vegetation at this scale and for that DVM modelling is needed". In the end of the paragraph we will add "As the only available climate projection at this resolution is a high-emission scenario, the simulated state at the end of the century will provide a message to society of what to expect and plan for if emissions continue to increase. We hypothesize that this state will show extensive changes that will present challenges for forestry, reindeer herding, forestry, tourism and nature conservation.". However, we do not agree that the objectives always should be used in the same order. You can build a good story in the discussion without following the objectives.

I would also suggest being consistent with presenting the results, especially for the evaluation of the modelled data. From my point of view, it would be useful either to use always the mean values for a certain period or the values of single years.

The reason why we have compiled the modelling results for 10-year periods, also when compared to inventoried or satellite-based products for single years, is that the random

nature of the disturbances in LPJ-GUESS otherwise could influence the results and comparisons. We should have explained this and we will add a statement in section 2.5.1.

Please add letters (a, b, c, …) to the single figures in combined figures.

We will add letters in the single figures where appropriate. Though note that we have headings for the columns and lines in which the figures are placed that may be more useful to specify the subfigures for the more complicated figures.

Maybe present the figures addressing the specific sites presented in Figure 1 always in the same order, e.g., from north to south.

We have chosen to present first mountain sites from north to south and then forest sites from north to south and we always use the same order.

I would suggest being consistent with the used units of the resolution or if the resolution is given in degree add the converted unit in brackets.

To be more consistent we will use km as main unit and for data originally in degree grid we will present a rough resolution in km. This is however not straight forward, as at these high latitudes the distance for a degree unit is 2-3 times larger in S-N direction than in W-E, depending on latitude (https://www.nhc.noaa.gov/gccalc.shtml).

Probably move parts of the 4ᵗʰ paragraph (Line 587 to 593) to the conclusion.

As also Leanne suggested to more clearly state the uncertainty in the conclusion, we will move these lines to the conclusions.

**Specific points:**

39-41: Sentence unclear.

For clarity we will revise the sentence to "For each degree of global average temperature increase the observed increase in Fennoscandia has been estimated to 2-3 degrees (Rantanen et al., 2022), and this relationship persists in future predictions (Ono et al., 2022).".

94: … here use the 3 km-scale climate projections…..

We will revise as suggested.

Figure 1a: The squares and points as well as the fonts should be made more legible.

By changing the tone of the background and colour of the text we will make the figure more legible.

Figure 1: please place the letters indicating the subfigures in the same position.

As there is a top-left inlet figure in b we placed the letter below. We will revise to get the letters in the most consistent way possible.

119-121: This sentence is for me not clear.

We will revise and divide into two sentences "The model simulates the development of cohorts, belonging to different plant functional types (PFTs), when competing for light, nitrogen and water in replicate patches (here set to 15 patches per simulated climate gridcell). Each patch represents an area of 1000 m$^2$.".

121: 1 km² (please be consistent in using the units).

It is well established that 1 km$^2$ means one square km, not 1000 square meters. 1000 m$^2$ is then not 1 km$^2$ but 0.001 km$^2$, we still think that it is most appropriate to express it as 1000 m$^2$ and prefer not to revise. The only place we use km$^2$ is in line 551, where it is an appropriate unit.

122-124: are the process descriptions related to cycling of water and carbon based on species composition?

Except for the soil, these processes are calculated at cohort level, we will add this information.

134: add the wetlands.

We agree that it is better to also mention the wetland PFTs in this first sentence of 2.2.1 and we will add that.

Table 2: Please specify the species more for each plant functional type, e.g., *Alnus* sp. could be a shrub or a tree. Add typical species for the PFTs C3G, pmoss, and C3G_wet.

We will revise the exemplified species as suggested.

Furthermore, you should remove the species *Vaccinium vitis-idaea* and *Vaccinium myrtillus* from this list as they show no clear habitat preference (occurence in two different PFTs). This could be important if the process descriptions are depending on the species composition.

Given that *V. vitis-idaea* is an evergreen species and *V. myrtillus* is deciduous these species still belong to different functional types for which they are common representative species We feel that these PFTs are sufficiently well distinguished in the model, and do not intend to revise this.

145-147: Please add the information you used for the fine-tuning of some model parameters.

As also pointed out by Leanne and Referee #2, we admit that it is not clearly expressed. We will revise lines 143-147 to "For the IBS plant functional type some parameters were changed according to Gustafsson et al. (2021) in an application for Abisko, Sweden. Their revision was made to reflect the fact that the global IBS PFT in Fennoscandia mainly represents mountain birch (*Betula pubescens* ssp. *tortuosa*). Details of the IBS parameterization are found in Table S2." and add this text to S2 "Using the default parameters, test runs with a sub sample of gridcells showed a substantial underestimation of deciduous broad-leafed forest adjacent to the mountains, where it is

represented by the Shade-Intolerant Broadleaved Summergreen tree plant functional type (IBS) in the simulations, when compared to the satellite-based products (see section 2.5.1). We therefore tested the IBS parameters from Gustafson et al. (2021), who had adjusted them to more specifically represent the small tree mountain birch (*Betula pubescens* ssp. *tortuosa*) and calibrated it to grow and compete as expected for the Abisko area (Gustafson personal communication). Simulations with these parameters on the other hand resulted in a too large extent of deciduous broad-leafed forest and to reduce the competitiveness of IBS the original values related to shade tolerance and turnover were used instead. For the same reason, we also adjusted the alphar and turnover_sap parameters." In Table S2 we will then also add the parameters that were not changed according to Gustafsson and add a column so that there will be "default 4.1", "Gustafsson et al. (2021)" and "used value".

169: Reference for the HCLIM38-ALADIN is missing and when you accessed the data.

We will add reference (Belušić et al., 2020), data access is described in the "Data availability" section.

170: Reference for the HCLIM38-AROME is missing and when you accessed the data.

We will add reference (Belušić et al., 2020), data access is described in the "Data availability" section.

181: Reference for the ALAatARO data is missing and when you accessed the data.

The ALAatARO dataset was produced in this study, which we will additionally clarify in the paper: In line 179 we will revise to "…when only ALADIN data were available, we first produced datasets such that the four…", and in line 181 to "(termed ALAatARO, 1985-2100)".

250-251: Which method was used for aggregating the satellite-based products to the 3 km resolution.

We used dominant class, we will add this information.

311: Please give the percentage for the UA and PA.

Thank you for pointing out this, we realize that this should be more specifically expressed. We will revise to "resulting in poor UA for broad-leaf deciduous shrubs (0.5%) and needle-leaf evergreen shrubs (0.0%), and poor PA for shrub tundra (0.2%).".

319-321: Did you use the average LAI for these periods or was it the change of LAI, PFTs respectively?

It was average, which we will specify.

381-382: "As the classification…." should be moved to the methods part.

We will refer to S5 where it is described instead of the details.

394: from which year is the vector-based map?

We will add the date (2021-09-09), both here and in S1.

472: From my point of view it is not the trend in grazing as the number of animals are constant over time and the authors simulate the grazing effect based on the loss of biomass. For me, if the authors talk about trends in grazing the authors should include the life stock units (LSUs).

We agree. The header will be revised to "Trends in potential reindeer grazing 2000-2100".

516-518: The thawing palsas were never mentioned before in the manuscript.

We will reformulate the sentence to make it clear that we bring up something new.

519: comma is missing after: e.g.,

Comma will be added.

Supplement S3 Line 40: (225 000 – 280 000)?

The missing zero will be added.

---

## Author Comment (AC2)

**RC2**: 'Comment on bg-2023-148', Anonymous Referee #2, 21 Nov 2023

**General comments**

**This study focuses on a subject of great scientific importance: changes in vegetation at high latitudes. The work presented is of high quality and based on state-of-the-art data and methods. The figures and the introduction and discussion sections are of good quality and in good shape for publication. The M&M and results sections could benefit from some changes in content and structure (see following paragraphs).**

Thank you for positive words and many useful comments, we will improve the M&A and discussion sections following your more specific comments.

**Specific comments**

**Introduction**

The introduction is clear, synthetic, and overall presents a sufficient number of scientific references. It describes the processes taking place in high-latitude regions, in particular the impacts of climate change and reindeer husbandry on the different ecosystems (boreal forest, tree-line and tundra). It presents the implication of vegetation dynamics modeling (DVM) and the added value enabled by scenario climate data at a 3km scale.

One introduction paragraph (line 54-61) should be moved to the Material and Method section, as it describes the study area and interrupts the flow developed in the introduction.

We will keep a smaller introduction of the region to introduce the next paragraph. But most of the text will be merged with the first paragraph of the M&M as suggested.

In addition, it would be interesting to present the initial hypotheses of the study, particularly with regard to the description made of the causes (societal and climatic) of change in plant communities.

As also the other reviewers have pointed out that task, objective and hypothesis should be clearly stated, we will revise the last section of the introduction. We will add this sentence after "which is particularly important in complex terrain.": "It is important to understand how climate change might affect the vegetation at this scale and for that DVM modelling is needed". In the end of the paragraph we will add "As the only available climate projection at this resolution is a high-emission scenario, the simulated state at the end of the century will provide a message to society of what to expect and plan for if emissions continue to increase. We hypothesize that this state will show extensive changes that will present challenges for forestry, reindeer herding, forestry, tourism and nature conservation."

Finally, it would be interesting to mention the LPJ-Guess model in the paragraph on DVM, in order to highlight its field of application.

We will mention LPJ-GUESS in the paragraph starting at line 78.

**M&M**

This section is well presented and provides a clear understanding of the data and studies used to complete this scientific production. More detail could be provided on certain paragraphs (see specific comments). More specifically, the choices made regarding data acquisition, interpolation type and parameter tuning must be sufficiently justified or discussed.

A comparative analysis of different climate (RCP) or socio-economic (SSP) scenarios would have been desirable. Thus, the authors of the study are invited to explain this choice of scenario.

With the addition to the last paragraph of the introduction suggested above, we don't think it would be needed to elaborate on the choice of scenario in M&M.

**Results**

This section provides well the overall results of the study with different type of figures. This section is quite dense and would benefit from structural reshaping, including topic sentences introducing most paragraphs.

We will not revise the general structure of the Results, but we will add introductory sentences to the sections where appropriate.

**Discussion & Conclusion**

I truly enjoyed reading the Discussion section which adequately explain the results and does not hide the limits of the method. I have very few comments on this section.

Thank you.

**Technical corrections**

38 « biological and societal significance » What do you mean by biological significance?

We will remove "biological".

82 « This information does not capture all local variation, especially in areas of complex terrain where altitudinal differences can be strongly underestimated » This sentence would require a reference.

We will add the Lind et al. (2020) reference also to this sentence.

91 « km-scale » Would be more rigorous to refer to 3km-scale throughout the text

The idea behind the term "km-scale" in climate modelling is to distinguish km-scale (implying 1-4 km) from coarser resolutions (>10 km). In other words, "km-scale" stands for the order of magnitude of 1 km (i.e. < 5 km), rather than the specific value of 1 km. In line 91, we will change it to "… first ever climate model projections at 3 km grid spacing…". We will still use the more general "km-scale" in more general contexts such as in the title and in line 94, and we will explain what the term means in connection to that line.

110 « The study area (shown as altitude from sea level (dark green) to 2000 m a.s.l. (yellow), the six focus "hotspot" areas (shaded squares and black text, see Figure S1 for detailed maps) » Instead of adding the description of symbols and colors in the text, use a legend directly in the figure with color (even for altitude) and symbol. What's more, the names don't appear clearly.

We will add a height legend and revise the colours so that the figure will be more legible.

127 « disturbances representing e.g. devastating pests or wind storms, occur randomly in each patch (return time set to 150 years in the presented simulations) » In natural/semi-natural ecosystem, disturbance does not occur randomly. Some ecosystems are more fire-prone (e.g. fire) depending on the season. You might want to discuss the "150yr return time at random" in the discussion, especially considering its huge significant impact in vegetation dynamics

We will discuss uncertainty related to disturbance return time in the vicinity of lines 605-610 in the discussion

139 « For fractions of land classified as peatland » Include peatland location in Fig. 1.a.

The peatland areas are too small and scattered to be shown in the overview of Figure 1a. The areas are shown in Figures 2, 4 and 6 as the peatland area is not dynamical but prescribed from this "PEATMAP". But due to the scale they can only be detected in Figure 6.

145 « A fine-tuning of some of the model's parameters was therefore done to get a better match against distribution maps from observations » Your parameter changes seem significant, especially for growing degrees days. Could you explain your choices in more detail? Does LPJ-GUESS present a poor calibration of this PFT in general or for this geographic zone in particular?

As also pointed out Referee #1 and Leanne, we admit that it is not clearly expressed. We will revise lines 143-147 to "For the IBS plant functional type some parameters were changed according to Gustafsson et al. (2021) in an application for Abisko, Sweden. Their revision was made to reflect the fact that the global IBS PFT in Fennoscandia mainly represents mountain birch (*Betula pubescens* ssp. *tortuosa*). Details of the IBS parameterization are found in Table S2." and add this text to S2 "Using the default parameters, test runs with a sub sample of gridcells showed a substantial underestimation of deciduous broad-leafed forest adjacent to the mountains, where it is represented by the Shade-Intolerant Broadleaved Summergreen tree plant functional type (IBS) in the simulations, when compared to the satellite-based products (see section 2.5.1). We therefore tested the IBS parameters from Gustafson et al. (2021), who had adjusted them to more specifically represent the small tree mountain birch (*Betula pubescens* ssp. *tortuosa*) and calibrated it to grow and compete as expected for the Abisko area (Gustafson personal communication). Simulations with these parameters on the other hand resulted in a too large extent of deciduous broad-leafed forest and to reduce the competitiveness of IBS the original values related to shade tolerance and turnover were used instead. For the same reason, we also adjusted the alphar and turnover_sap parameters." In Table S2 we will then also add the parameters that were not changed according to Gustafsson and add a column so that there will be "default 4.1", "Gustafsson et al. (2021)" and "used value".

150 « Grazing/browsing was simulated by removing a fraction of leaf biomass. » Provide a reference to support the assumption that grazing/browsing only/mainly affects leaf biomass. Some could argue that plant survival or growth could also be directly affected.

LPJ-GUESS has only stem, leaf and fine-root compartments, grass PFTs only leaf and fine-roots. We have not found any study of what proportion of the consumed biomass that is twigs/branches. We will acknowledge that we may have underestimated the effect of reindeer browsing in the discussion paragraph starting at line 548, based references that show the height differences of the vegetation between open and exclosure sites. We will also add a comment that there are rather small differences in simulations with or without reindeer presence in section 3.3.1.

200 « The RCP8.5 scenario used was the first dataset produced at this high resolution for the entire region » The first and only one? Please indicate why using this particular scenario and not propose a comparative analysis with Shared Socioeconomic Pathway to get a nuanced vision of vegetation dynamics in the future.

As stated above, we will state already in the introduction that there indeed really was only this scenario available at this resolution.

210 « 5 arc seconds (10 km) » Please use consistent unit for resolution (km preferred)

To be more consistent we will use km as main unit and for data originally in degrees we will present rough resolution in km. It is however not straight forward, as at these high latitudes the distance for a degree unit is 2-3 times larger in S-N direction than in W-E, depending on latitude (https://www.nhc.noaa.gov/gccalc.shtml).

211 « had been interpolated to 3 km resolution » What type of interpolation?

"interpolated" is not the appropriate expression here as they were just taken from the relevant 10 km data cell. We will just skip the expression after the parenthesis.

225 « downscale the 50 km data for the 1900-1986 period. After 2051, the 0.5° resolution Lamarque et al. (2011) dataset was used, which is standard for LPJ-GUESS. » How the final resolution of 3km is reached?

The Ndep data were simply at a coarser resolution. We are aware that this could be a potential problem, but we have done a sensitivity test using the coarser 0.5° Ndep data and compared it to simulations using the high-resolution data, finding very small differences in simulated vegetation composition. We will add this information but will not provide specific results.

226 « 0.5° » require consistent unit

To be more consistent we will use km as main unit and for data originally in degrees we will present rough resolution in km. It is however not straight forward, as at these high latitudes the distance for a degree unit is 2-3 times larger in S-N direction than in W-E, depending on latitude (https://www.nhc.noaa.gov/gccalc.shtml).

267 « to convert modelled total biomass to above ground biomass we assumed a factor of 0.85 based on earlier estimates » Root:shoot ratios could be very variable for foret tree species. For instance, in (Huttunen et al. 2013) they estimated that for silver birch seedlings, the ratio between BGB and AGB could vary from 0.35 to 1.2 under different conditions and would be around 0.55 without treatment. Even though you consider here adult trees, the approximation of 0.85 should be discussed.

Huttunen, Liisa & Ayres, Matthew & Niemelä, Pekka & Heiska, Susanne & Tegelberg, Riitta & Rousi, Matti & Kellomäki, Seppo. (2013). Interactive effects of defoliation and climate change on compensatory growth of silver birch seedlings. Silva Fennica. 47. e964. 10.14214/sf.964.

We will add the range found by Huttunen et al. (2013) to the range by Johansson (2007) when showing uncertainty of the 0.85 value.

299 « Simulated_LAI = 0.78 × SURFEX_LAI – 1.99, r2 = 0.59 » You might consider providing the results with an intercept of 0. In addition, even if an r² of 0.59 can be considered satisfactory, you should explain where and why the simulated LAI differs from the SURFEX data.

As there is such a large intercept (-1.99), we don't think it would be useful to show statistics for a regression through zero. We will comment on this large intercept and reasons for the big difference.

340 « Figure 3» Compared to the high quality of the other figures, this one could be reshaped and made more clear

We will make an overhaul of the figure to improve the quality.

381 « As the classification is based on LAI, bare rock was set for LAI 0.01- 0.001 and permanent snow/ice < 0.001, indicating that plants have the potential to grow there » Unclear and might be put in M&M section

Instead of the details given between the commas in the sentence of lines 381-382, we will refer to supplement S5 where it is described: "…based on LAI (see S5) it indicates that plants…".

385 « Figure 5 » This figure is really interesting and well designed, congrats. One thing, to be more rigorous, the 3 rows should me made from equivalent latitude band sample ( not 0.2, 0.15 and 0.1 °)

Thank you for the appreciation. Again, the bands should not be equally wide as the number of gridcells per latitude degree is different. Now the area of the bands has an area of 10000, 8500 and 7300 km$^2$ from north to south. We will make a revision of the figure for consistency.

442 « It should be kept in mind that the data obtained from the Analysis Portal relies on what has been reported by a large community of public and professional naturalists, which means that biases can exist e.g., depending on the specific biological interests of rapporteurs visiting the different areas. » this sentence might be preferably put in Discussion section and more developed. Indeed, this section aim at providing an overview of selected hotspot diversity but in my opinion the method for this purpose should be more discussed.

We will move this statement to the discussion line 533. We think that discussion of the details of methods for collecting this type of data would take too much focus from the main aspects of the article.

478 « Figure 8 » The colour legend might be redefined to better show region with low and high change of consumption. In addition, it could be interesting to plot the relative change (percentage) instead of the absolute one which poorly reflects effect on vegetation.

We will try to find a better colour scheme. To show relative change will not be possible as many grid cells are at zero in present period.

501 « dramatic » This term has a negative connotation, but the consequences of such a change in vegetation are complex and not entirely negative, depending on the process under consideration.

We will change to a more neutral word: "extensive".

544 « This is because many of the alpine species in the hotspots areas that are not listed today will be threatened as warming continues. » This need a literature reference.

We will add a reference.

---

## Author Comment (AC3)

- **CC1**: 'Comment on bg-2023-148', Leanne van der Kuijl, 11 Nov 2023

*This review was prepared as part of graduate program course work at Wageningen University. The review was not solicited by the journal, but it might be of some use.*

Your review will certainly be of substantial use, and we thank you for careful reading and many good comments and suggestions.

This paper uses high resolution simulations to assess changes in vegetation composition, biodiversity, and available reindeer forage in the 21st century due to climate change and potential reindeer grazing in the Fennoscandian region. The authors parameterized the LPJ-GUESS dynamic vegetation model with the PFTs found in the region and added a reindeer grazing module. The model was forced using a downscaled high-emission climate scenario (RCP8.5). Validation on local scale was done using vegetation inventories and on regional scale using remote sensing maps. The vegetation shifts were analyzed in more detail in six diverse "hotspot" areas. The results show dramatic south to north and low to high altitude shifts in vegetation zones, that accelerate towards the end of the 21st century. Potential reindeer grazing ground will also shift to the north based on the availability of suitable forage, but other factors resulting from climate change are likely to impact reindeer husbandry more. The authors conclude scenario-based research is needed to better assess the vegetation changes in the future and their uncertainty.

As far as I know this paper is the first to attempt to integrate the effect of reindeer grazing, which is an important landscape forming factor in Fennoscandia, into a dynamical vegetation model. This paper is also the first the use a very high-resolution climate scenario, which better captures the local variation in complex terrain, on the entire Fennoscandian boreal and Oroarctic region. The research is necessary and valuable in showing the severity of the consequences of climate change for nature conservation and for the conservation of the indigenous culture of the region (reindeer husbandry).

The study seems well designed, despite only using one climate scenario, and the methodology, calibration and validation seem to follow generally accepted protocols for dynamic vegetation modeling experiments. Additionally, the figures in the paper visualize the results well. However, the methodology is hard to read, and the introduction is missing some key arguments as to why the study is important. I also would like to see some processes and effects that were not considered in the model added to the discussion, but most importantly, I feel like the uncertainty of this study needs to be stated more clearly in the summary and the conclusion.

We thank you for your positive words and hope that the problems you identified will be solved when we have dealt with the specific comments.

In my opinion this paper will be suitable for publication in Biogeosciences after moderate revision.

**Major arguments**

The model used in this study is only forced using one (extreme) climate scenario and there is a lot of (unquantified) uncertainty in the results. The authors do already state in the discussion that this uncertainty exists and could not be solved due to computational restrictions and because there currently are no different climate scenarios at the high resolution of this study available. Despite this, the study is still a relevant and necessary first step. However, I do feel it is very important to mention the uncertainty of this study more clearly in both the conclusion and the summary to avoid sensationalizing the results. If I were to read only the title, summary, and conclusion of this paper, I would not know this paper only describes general trends. Especially in the summary that states this region **will** be completely covered by forests at the end of the 21$^{st}$ century.

The conclusion and summary will be revised to clearly state that this is a high-emission scenario. Some of the uncertainty discussion will also be moved to the conclusion.

In my opinion both the methodology and introduction require some additions to make it easier for people to find the relevant information and to clearly express the relevance and necessity of this study:

The modelling methodology is in my eyes poorly described and rather wordy. It is unclear what model is used and where a detailed description of this model can be found. I had to consult additional literature to try and figure out what model was used (Gustafson et al., 2021; Smith et al., 2001; Miller and Smith, 2012; Smith et al., 2014) and I am still not sure, because Gustafson at al. (2021) only mentions LPJ-GUESS v4.0. This could be remedied by adding a schematic overview of the model highlighting which parts of the model are new (reindeer module and some PFTs) and where the detailed descriptions for the old parts can be found.

We will clarify what model was used and where it is described in detail. We don't intend to add a schematic overview, but do describe clearly the new features and parameterizations you mention.

Additionally, it is unclear which data was used to calibrate which parameters of the model. For example, lines 143-147 on page 3 mention test runs and fine-tuning to get a better match against distribution maps from observations, but it does not mention what maps. I feel adding a (supplementary) table of which data sets were used to calibrate which parameters (covering what time period) would solve this problem and prevent people having to comb through the entire paper to figure out what was done.

As also pointed out Referee #1 and #2, we admit that it is not clearly expressed. We will revise lines 143-147 to "For the IBS plant functional type some parameters were changed according to Gustafsson et al. (2021) in an application for Abisko, Sweden. Their revision was made to reflect the fact that the global IBS PFT in Fennoscandia mainly represents mountain birch (*Betula pubescens* ssp. *tortuosa*). Details of the IBS parameterization are found in Table S2." and add this text to S2 "Using the default parameters, test runs with a sub sample of gridcells showed a substantial underestimation of deciduous broad-leafed forest adjacent to the mountains, where it is represented by the Shade-Intolerant Broadleaved Summergreen tree plant functional type (IBS) in the simulations, when compared to the satellite-based products (see section 2.5.1). We therefore tested the IBS parameters from Gustafson et al. (2021), who had adjusted them to more specifically represent the small tree mountain birch

(*Betula pubescens* ssp. *tortuosa*) and calibrated it to grow and compete as expected for the Abisko area (Gustafson personal communication). Simulations with these parameters on the other hand resulted in a too large extent of deciduous broad-leafed forest and to reduce the competitiveness of IBS the original values related to shade tolerance and turnover were used instead. For the same reason, we also adjusted the alphar and turnover_sap parameters." In Table S2 we will then also add the parameters that were not changed according to Gustafsson and add a column so that there will be "default 4.1", "Gustafsson et al. (2021)" and "used value". Except for this there were no calibration done, and we don't think any table more than Table S2 will be needed.

In the introduction the paper is vague on what the consequences of a shift in vegetation composition might be and how reindeer grazing might affect and be affected by this shift. It just mentions increasing pressure on both ecosystems, holding species of great ecological, biological, and societal significance, and societies in the area. It seems to me that the consequences of such a shift in vegetation due to climate change are the most important reason for doing this research. I feel it would be beneficial to be more specific or perhaps add some examples to get the message across better. Stark et al. (2023) gives a lot of information about effects of reindeer on the ecosystem. It is also worth mentioning how culturally important reindeer husbandry is for the indigenous people of the region.

We will explain consequences of climate-change related shifts in vegetation composition to show that this research is important using the Stark reference and others, and will mention the cultural importance for the Sami. We will also more clearly express the task, objective and hypothesis of the study in the end of the introduction.

Building further on this point: The summary and conclusion mention more about the consequences than the introduction does. The conclusion for example mentions implications for recreation when this is not mentioned anywhere else. These extra points should be moved to or already be mentioned in the introduction.

As mentioned above we will mention aspects such as recreation in the introduction.

I would also like to see more discussion about processes and effects that were not considered in the model. The following points being most important:

Currently only one sentence in the paper (line 607) mentions seed dispersal capacity (and fire disturbance) as factors that may restrict vegetation expansion, particularly for predicted shrub expansion on non-shrub tundra. However, as mentioned by Gustafson et al. (2021) dispersal limitations are likely to cause lags in range shifts on larger spatial scales (Rees et al., 2020; Brown et al., 2018). Models that do account for seed dispersal limitations generally predict slower latitudinal tree migration than models driven solely by climate like LPJ-GUESS (Epstein et al., 2007). This warrants further explanation in the paper.

We will extend the discussion with possible consequences of not accounting for seed dispersal rate, including some of the suggested references as well as Zani et al (2022, https://doi.org/10.5194/gmd-15-4913-2022).

This paper discusses the direct effects reindeer grazing, browsing, and trampling might have on the vegetation and how the change in vegetation due to climate change might affect the food supply for the reindeer, but it does not discuss the effect reindeer might have on the climate and thus indirectly on the vegetation as well. Recently Holmgren et al. (2023) found that a high amount of reindeer summer trampling in low peatland areas may result in increased summer warming and decreased winter cooling enhancing permafrost degradation in these areas. On the other hand, in higher areas intense browsing and nutrient addition from reindeer may mitigate some climate warming effects (Macias-Fauria et al., 2020; Malhi et al., 2022). This uncertainty should be added to the discussion.

We will add a discussion about possible climate feedback from reindeer grazing.

Furthermore, it should be noted that the reindeer exclosure sites that are used to validate the reindeer grazing module likely do not represent the natural situation without grazing (Stark et al., 2023).

We will acknowledge this.

Soil vertical and horizontal movement caused by frost, and amelioration of such effects in the warmer future climate are not accounted for in the LPJ-GUESS model. These processes could affect survival and competition among the plant functional types, especially in the seedling stage when plants are most vulnerable to mechanical disturbance (Gustafson et al., 2021). This should also be added to the discussion, if relevant on high resolution larger scale.

There are a lot of simplifications when it comes to considering processes that affect the establishment of new plants; the mentioned soil processes, seed predation, plant browsing and killing by smaller animals like rodents and hares, deceases, snow damage, moose, etcetera. We will mention this in general terms, but an extended discussion of all these aspects would take too much space.

**Minor arguments**

1. Page 3, lines 104-105: Why was this study area chosen? The summary, introduction and title make it feel like the entire Fennoscandian area will be studied, when it is not. I feel like some explanation is necessary here.

In the introduction we clearly state that we address the "Fennoscandian boreal and Oroarctic region" (line 54, 89, 96). That the study is also restricted to the area used for reindeer herding is explained in M&M, but we will add this information also to the introduction (line 96). The eastern border in Finland, where a small part that could have been included is outside the assessed area, is due to the fact that we for computational reasons wanted to restrict the size of the complete 1985-2100 NetCDF climate files to less than 32 GB per climate variable, we will include this information in section 2.3.

2. Page 5, line 128: The return time of patch destroying disturbances (e.g., devastating pests or windstorms) is set to 150 years. What is this based on? Does this also include fires? This requires some explanation.

Fire is simulated separately, we will add information on that. 150 years is the default return time for arctic simulations in LPJ-GUESS 4.1, we will add that information as well.

3. Page 6, lines 158-159: Barthelemy et al. (2018) says that nitrogen in the form of urea can easily be taken up by plants directly (as in without transforming it, not as in how quickly). The way it is currently formulated in this paper makes it feel as if the assumption is that the nitrogen is immediately (time) taken up by the plants. Barthelemy et al. (2018) does not seem to state how quickly this happens other than that it happens 'rapidly'. Urine is also a very local phenomenon, but for the model it is taken up by plants in the entire patch.

We agree that "directly" is a misleading choice of word and will change to "rapidly" both here and in the supplement (line 56).

4. Page 7/8, section 2.4: Is the nitrogen data at 12 km resolution detailed enough for this study with a resolution of 3 km? This is not mentioned anywhere yet.

We have done a sensitivity test using the more coarsely resolved 0.5° Ndep data and compared it to simulations using the high-resolution data, and found very small differences in simulated vegetation composition. We will add this information but not provide specific results.

5. Page 15/16, section 3.2.2: I wonder what the effect is of leaving out the wetland areas in the calculation of the Shannon Diversity Index on the comparison between the different time periods. Specifically in Muddus, which has a lot of wetland area in the simulated data, but barely any in the satellite data. This should be mentioned.

We explain in M&M that we exclude cells that do not have a dynamic vegetation class like water and wetlands, but we will add this information also to the table text of Table 3. We will also reformulate the sentence of line 416-417 to "It should also be noted that the bog class, which covers a large fraction in Muddus, is excluded from the calculation of $D$ as it is prescribed and not dynamic. Including it would increase $D$ but we would not be able to assess the change over time.".

6. Page 22, line 556: I feel it would be better to mention just how much larger the reindeer herd size is in summer (60% larger after calving and before slaughter, Definitions - Sámi Parliament (sametinget.se)), because this could result in a significant underestimation of the grazing.

We can't see where the number 60% comes from, according to https://sametinget.se/statistik/renslakt (in Swedish) the number of slaughtered animals is 40 000 – 75 000 reindeers, which rather means that the summer population is about 25% larger. The animals that are slaughtered are also often calves that eat less. We already acknowledge that there is an underestimation in the current formulation. We don't intend to revise.

7. Supplement page 4 (S3), line 38: Please add and explanation as to why the Swedish reindeer population is (assumed to be) representative for the entire study area.

We will add the information that we will use Sweden as an example for verification. We write in line 105 in the main text that the focus of the study will be on Swedish ecosystems.

8. Supplement page 4 (S3), lines 53-59: This does not explain why 35% was used. Ferraro et al. (2022) assumes that 38% of the daily nitrogen consumed by reindeer is assimilated into its body mass and 62% is defecated (not all as urine), instead of the other way around. McEwan and Whitehead (1970) show that nitrogen divers a lot depending on the age and sex of the reindeer and on the season. Nitrogen retention during the second growth phase (14-24 months) was on average 35% in reindeer (page 909, table IV). The remaining 65% was presumed to be urine. I am presuming the 35% used in this study is based on this, but I am not entirely convinced this is representative for the entire reindeer population. Please explain in more detail.

It is not the other way around, entering the harvest pool is the same as being assimilated in body mass, e.g. being removed from circulation in the ecosystem. To go into detail of the proportion of different growth phases would be overly complicated for this simple model. Based on your recommendations, we will check the text carefully to improve clarity and add a reference.

**Minor issues:**

Main article

Page 1, lines 30-32: "Simulated … grazing ground." This sentence is unclear and needs to be rewritten.

For clarity, we will split the sentence into two: "Simulated potential for reindeer grazing indicates latitudinal differences, with higher potential in the south in the current climate. In the future these differences will diminish, as the potential will increase in the north, especially for the summer grazing grounds."

Page 2, line 37: Reference?

We will add this reference: Dobrowski et al., 2021. Protected-area targets could be undermined by climate change-driven shifts in ecoregions and biomes, Communications Earth & Environment, 2, https://doi.org/10.1038/s43247-021-00270-z.

Page 3, line 87: "have" should be "has".

"Data" is always plural (datum is the singular form), we will not revise.

Page 3, line 95: Perhaps add a more recent reference for the "state-of-the-art DMV" here like Gustafson et al. (2021). 2014 is not state-of-the-art.

We will add Lindeskog et al. 2021.

Page 6, line 175: "(see below)" What does this reference? There is nothing below.

We will remove this reference as it is not needed.

Page 7, line 182: "two" methods" are announced here, but only one of those methods is explained in the same paragraph and the other in the next paragraph.

We will place a colon after "using two methods" and then start a new paragraph, so that each method has a separate paragraph.

Page 8, line 246: "also" before "converted" should be left out.

We will revise the entire sentence to "Further, modelled LAIs for the specific PFTs within a grid cell were used to determine a vegetation class for comparison to two remote-sensing based vegetation products: the land cover of northern Eurasia (GLCE) based on SPOT 4 at 1 km resolution (Bartalev et al., 2003) and the CLC2018 Corine land-cover dataset (Corine) based on Sentinel 2 at 100 m resolution (https://land.copernicus.eu/pan-european/corine-land-cover/clc2018).".

Page 13, line 363: "were" should be "was".

We will revise.

Page 21, lines 524-525: "which is not far from the current trajectory" reference?

We will revise the entire sentence: "Thus, with a continued warming of up to 5 K by the end of the century, which is not totally unlikely as the world is currently on track for a 2.9 K increase in global warming (Ref to UN Emissions Gap Report 2023, https://www.unep.org/resources/emissions-gap-report-2023) and given the Arctic warming amplification (Rantanen et al., 2022), the Fennoscandian vegetation is likely to undergo a rapid shift.".

Page 22, line 554: either "since" or "as" needs to be left out.

"since" will be removed.

Page 29, line 805: This references the preprint of this paper, not the published version.

It will be fixed.

*Supplement*

Supplement page 4, line 34: "reduce" should be "reduces".

We will revise accordingly.

Supplement page 4, line 39-40: add the specific webpage where the numbers used can be found.

The webpage we are referring to specifically provides these numbers, though it is in Swedish.

Supplement page 4, line 40: "280 00" should be "280 000".

The missing zero will be added.

Supplement page 4, line 41: "eat" should be "eats".

We will revise accordingly.

Supplement page 4, line 41: I think "path" is supposed to be "patch" here.

Correct, we will revise.

Supplement page 4, line 46: missing "the" between "large" and "fraction".

We will revise accordingly.

Supplement page 4, line 47: add "is" between "consumed" and "relative" and perhaps add "where" between "and" and "herbivore_int" to make the sentence clearer.

We will revise accordingly.

Supplement page 4, line 55: missing "a" between "is" and "functionality".

We will revise accordingly.

Supplement page 4, line 55-56: add a reference for the assumption that N leaving the herbivore as urine is directly taken up by the plants.

We will add a reference, see also comment to "Minor argument nr 8".

Supplement page 7, line 112: The Bartalev et al. (2003) reference is missing in the reference list.

Supplement page 8, line 119: Babst et al. (2014) is not referenced in the text.

By mistake Babst et al. (2014) was inserted in the reference list instead of Bartalev et al. (2003). The mistake will be corrected.

Supplement page 18-19: What do the colours in this figure mean?

We will add an explanation of the colours

**References**

Barthelemy, H., Stark, S., Michelsen, A., & Olofsson, J. (2017). Urine is an important nitrogen source for plants irrespective of vegetation composition in an Arctic tundra: Insights from a 15 N-enriched urea tracer experiment. Journal of Ecology, 106(1), 367–378. https://doi.org/10.1111/1365-2745.12820

Brown, C. M., Dufour-Tremblay, G., Jameson, R. G., Mamet, S. D., Trant, A. J., Walker, X. J., Boudreau, S., Harper, K. A., Henry, G. H. R., Hermanutz, L., Hofgaard, A., Исаева, Л. Г., Kershaw, G. P., & Johnstone, J. F. (2018). Reproduction as a bottleneck to treeline advance across the circumarctic forest tundra ecotone. Ecography, 42(1), 137–147. https://doi.org/10.1111/ecog.03733

Epstein, H. E., Qin, Y., Kaplan, J. O., & Lischke, H. (2007). Simulating future changes in Arctic and Subarctic vegetation. Computing in Science and Engineering, 9(4), 12–23. https://doi.org/10.1109/mcse.2007.84

Ferraro, K. M., Schmitz, O. J., & McCary, M. A. (2021). Effects of ungulate density and sociality on landscape heterogeneity: a mechanistic modeling approach. Ecography, 2022(2). https://doi.org/10.1111/ecog.06039

Gustafson, A. F., Gustafson, A. F., Björk, R. G., Olin, S., Smith, B., Björk, R. G., Olin, S., Smith, B., & Smith, B. (2021). Nitrogen restricts future sub-arctic treeline advance in an individual-based dynamic vegetation model. Biogeosciences, 18(23), 6329–6347. https://doi.org/10.5194/bg-18-6329-2021

Holmgren, M., Groten, F., Carracedo, M. R., Vink, S., & Limpens, J. (2023). Rewilding risks for Peatland permafrost. Ecosystems. https://doi.org/10.1007/s10021-023-00865-x

Macias-Fauria, M., Jepson, P., Zimov, N., & Malhi, Y. (2020). Pleistocene Arctic megafaunal ecological engineering as a natural climate solution? Philosophical Transactions of the Royal Society B, 375(1794), 20190122. https://doi.org/10.1098/rstb.2019.0122

Malhi, Y., Lander, T. A., Roux, E. L., Stevens, N., Macias-Fauria, M., Wedding, L. M., Girardin, C., Kristensen, J. A., Sandom, C. J., Evans, T., Svenning, J., & Canney, S. M. (2022). The role of large wild animals in climate change mitigation and adaptation. Current Biology, 32(4), R181–R196. https://doi.org/10.1016/j.cub.2022.01.041

McEwan, E. H., & Whitehead, P. E. (1970). Seasonal changes in the energy and nitrogen intake in Reindeer and Caribou. Canadian Journal of Zoology, 48(5), 905–913. https://doi.org/10.1139/z70-164

Miller, P. A., & Smith, B. (2012). Modelling Tundra vegetation response to recent Arctic warming. AMBIO: A Journal of the Human Environment, 41(S3), 281–291. https://doi.org/10.1007/s13280-012-0306-1

Rees, G., Hofgaard, A., Boudreau, S., Cairns, D. M., Harper, K. A., Mamet, S. D., Mathisen, I. E., Swirad, Z., & Tutubalina, O. (2020). Is subarctic forest advance able to keep pace with climate change? Global Change Biology, 26(7), 3965–3977. https://doi.org/10.1111/gcb.15113

Smith, B., Prentice, I. C., & Sykes, M. T. (2001). Representation of vegetation dynamics in the modelling of terrestrial ecosystems: comparing two contrasting approaches within European climate space. Global Ecology and Biogeography, 10(6), 621–637. https://doi.org/10.1046/j.1466-822x.2001.t01-1-00256.x

Smith, B., Wårlind, D., Arneth, A., Hickler, T., Leadley, P., Siltberg, J., & Zaehle, S. (2014). Implications of incorporating N cycling and N limitations on primary production in an individual-based dynamic vegetation model. Biogeosciences, 11(7), 2027–2054. https://doi.org/10.5194/bg-11-2027-2014

Stark, S., Horstkotte, T., Kumpula, J., Olofsson, J., Tømmervik, H., & Turunen, M. (2023). The ecosystem effects of reindeer (Rangifer tarandus) in northern Fennoscandia: past, present and future. Perspectives in Plant Ecology, Evolution and Systematics, 58, 125716. https://doi.org/10.1016/j.ppees.2022.125716

---

## Author Response (AR1)

Here follows action taken to the reviewer comments with line references to the non-marked-up version of the document

- **RC1**: 'Comment on bg-2023-148', Anonymous Referee #1, 06 Nov 2023
  **Common points:**
  The paper aims to model the vegetation development until 2100 in the Fennoscandinavian to Oroarctic environment due to climate change in a high resolution, including the impact of reindeer grazing/trampling and the effects on biodiversity. The aims of the paper fit well to the scopes of the journal.
  We thank you for careful reading and many useful comments that will improve the paper.

  In my opinion the resolution is not high. High resolution should be 100 m to 10 m. The data presented here have a resolution of 3 km. Furthermore, the study area extends from the boreal forest to the tundra. Therefore, I would suggest adjusting the title.
  What high resolution means depends on what field you are working in. In the context of regional climate modelling, as well as large-scale combined climate-vegetation modelling, 3 km represents a higher resolution than in most other studies. However, we agree that you have a point about the scale and have revise the title to emphasize that simulations were done over a large area (1-2).

  The second paragraph of the introduction should be moved to the description of the study area.
  The paragraph has been moved (from 55 to 108-115).

  In the last paragraph of the introduction (Line 93 to 100) it could improve the manuscript (structure) if the task and the objectives are clearly stated. The objectives should then be always used in the same order in the other sections of the paper (methods, results, and discussion).
  As also the other reviewers have pointed out that task, objective and hypothesis should be clearly stated, we will revise the last section of the introduction. We have added a clarification of the task (91-92) and specified the objectives (100-103). However, we do not agree that the objectives always should be used in the same order. You can build a good story in the discussion without following the objectives.

  I would also suggest being consistent with presenting the results, especially for the evaluation of the modelled data. From my point of view, it would be useful either to use always the mean values for a certain period or the values of single years.
  The reason why we have compiled the modelling results for 10-year periods, also when compared to inventoried or satellite-based products for single years, is that the random nature of the disturbances in LPJ-GUESS otherwise could influence the results and comparisons. We should have explained this and we have added a statement in 2.5 (270-272).

  Please add letters (a, b, c, …) to the single figures in combined figures.
  We have added letters in the single figures (Fig 4, Fig 5, Fig 7, Fig 8, Fig 9). For Fig 6, we have headings for the columns and lines in which the figure is placed that we think are more useful to specify the subfigures in this complicated figure.

  Maybe present the figures addressing the specific sites presented in Figure 1 always in the same order, e.g., from north to south.
  We have chosen to present first mountain sites from north to south and then forest sites from north to south and we always use the same order.

I would suggest being consistent with the used units of the resolution or if the resolution is given in degree add the converted unit in brackets.
This is not straight forward, as at these high latitudes the distance for a degree unit is 2-3 times larger in S-N direction than in W-E, depending on latitude (https://www.nhc.noaa.gov/gccalc.shtml). To be more consistent we have used km as main unit and for data originally in degree grid we have present the resolution in km in S-N and W-E in parenthesis (154-155, 235, 251).

Probably move parts of the 4th paragraph (Line 587 to 593) to the conclusion.
As also Leanne suggested to more clearly state the uncertainty in the conclusion, we have moved these lines to the conclusions (from 633 to 689-696).

**Specific points:**
39-41: Sentence unclear.
The sentence has been clarified (39-41).

94: … here use the 3 km-scale climate projections…..
We have revised the section above to clearly show what we mean by km-scale (88-91)

Figure 1a: The squares and points as well as the fonts should be made more legible.
We have increased the font of the text and added shading to make it stand out better. The edge of the dots and squares has been made thicker and the shading of the squares removed (Fig 1a).

Figure 1: please place the letters indicating the subfigures in the same position.
The letters have been placed in the same position, to enable that, the inlet in b was moved to the right (Fig 1).

119-121: This sentence is for me not clear.
For clarity we have divided the sentence into two sentences (131-134).

121: 1 km$^2$ (please be consistent in using the units).
It is well established that 1 km$^2$ means one square km, not 1000 square meters. 1000 m$^2$ is then not 1 km$^2$ but 0.001 km$^2$, we still think that it is most appropriate to express it as 1000 m$^2$ and have not revised.

122-124: are the process descriptions related to cycling of water and carbon based on species composition?
Except for the soil, these processes are calculated at cohort level, we have added this information (137-138).

134: add the wetlands.
We agree that it is better to also mention the wetland PFTs in this first sentence of 2.2.1 and we have revised accordingly (147-149).

Table 2: Please specify the species more for each plant functional type, e.g., *Alnus* sp. could be a shrub or a tree. Add typical species for the PFTs C3G, pmoss, and C3G_wet.
We have added examples of species for *Alnus*, *Salix* (in three categories) and *Sphagnum*, the grass is a very diverse group and we have just added the family name (Tab 2).

Furthermore, you should remove the species *Vaccinium vitis-idaea* and *Vaccinium myrtillus* from this list as they show no clear habitat preference (occurence in two different PFTs). This could be important if the process descriptions are depending on the species composition.

Given that *V. vitis-idaea* is an evergreen species and *V. myrtillus* is deciduous these species still belong to different functional types for which they are common representative species We feel that these PFTs are sufficiently well distinguished in the model, and have not revised this.

145-147: Please add the information you used for the fine-tuning of some model parameters.

As also pointed out by Leanne and Referee #2, we admit that it is not clearly expressed. We have revised the text (156-159), and added a new section in S2 (S27-36). In Table S2 we added the parameters that were not changed according to Gustafsson and added a column so that there is "default 4.1", "Gustafsson et al. (2021)" and "used value" (Tab S2).

169: Reference for the HCLIM38-ALADIN is missing and when you accessed the data.

We have added the reference (Belušić et al., 2020) (190), data access is described in the "Data availability" section.

170: Reference for the HCLIM38-AROME is missing and when you accessed the data.

We have added the reference (Belušić et al., 2020) (191), data access is described in the "Data availability" section.

181: Reference for the ALAatARO data is missing and when you accessed the data.

The ALAatARO dataset was produced in this study, which we have additionally clarified in the paper (202, 204).

250-251: Which method was used for aggregating the satellite-based products to the 3 km resolution.

We used dominant class, we have added this information (281-282).

311: Please give the percentage for the UA and PA.

Thank you for pointing out this, we realize that this should be more specifically expressed have revise the whole sentence (347-348).

319-321: Did you use the average LAI for these periods or was it the change of LAI, PFTs respectively?

It was average, which we now specify (357).

381-382: "As the classification…." should be moved to the methods part.

We have now referred to S5, where it is described, instead of the details (421-422).

394: from which year is the vector-based map?

We have added the date when we accessed the data, both here (433-434) and in S1 (S23).

472: From my point of view it is not the trend in grazing as the number of animals are constant over time and the authors simulate the grazing effect based on the loss of biomass. For me, if the authors talk about trends in grazing the authors should include the life stock units (LSUs).

We agree. The header has been revised (511), also including RC2's suggestion of better introductions to the Results sections.

516-518: The thawing palsas were never mentioned before in the manuscript.
We have decided to remove this section as it is not something we address in this study (was at 556).

519: comma is missing after: e.g.,
Comma has been added (557).

Supplement S3 Line 40: (225 000 – 280 000)?
The missing zero has been added (S54).

- **CC1**: 'Comment on bg-2023-148', Leanne van der Kuijl, 11 Nov 2023
  ***This review was prepared as part of graduate program course work at Wageningen University. The review was not solicited by the journal, but it might be of some use.***
  Your review will certainly be of substantial use, and we thank you for careful reading and many good comments and suggestions.

This paper uses high resolution simulations to assess changes in vegetation composition, biodiversity, and available reindeer forage in the 21$^{st}$ century due to climate change and potential reindeer grazing in the Fennoscandian region. The authors parameterized the LPJ-GUESS dynamic vegetation model with the PFTs found in the region and added a reindeer grazing module. The model was forced using a downscaled high-emission climate scenario (RCP8.5). Validation on local scale was done using vegetation inventories and on regional scale using remote sensing maps. The vegetation shifts were analyzed in more detail in six diverse "hotspot" areas. The results show dramatic south to north and low to high altitude shifts in vegetation zones, that accelerate towards the end of the 21$^{st}$ century. Potential reindeer grazing ground will also shift to the north based on the availability of suitable forage, but other factors resulting from climate change are likely to impact reindeer husbandry more. The authors conclude scenario-based research is needed to better assess the vegetation changes in the future and their uncertainty.

As far as I know this paper is the first to attempt to integrate the effect of reindeer grazing, which is an important landscape forming factor in Fennoscandia, into a dynamical vegetation model. This paper is also the first the use a very high-resolution climate scenario, which better captures the local variation in complex terrain, on the entire Fennoscandian boreal and Oroarctic region. The research is necessary and valuable in showing the severity of the consequences of climate change for nature conservation and for the conservation of the indigenous culture of the region (reindeer husbandry).

The study seems well designed, despite only using one climate scenario, and the methodology, calibration and validation seem to follow generally accepted protocols for dynamic vegetation modeling experiments. Additionally, the figures in the paper visualize the results well. However, the methodology is hard to read, and the introduction is missing some key arguments as to why the study is important. I also would like to see some processes and effects that were not considered in the model added to the

discussion, but most importantly, I feel like the uncertainty of this study needs to be stated more clearly in the summary and the conclusion.

We thank you for your positive words and hope that the problems you identified will be solved when we have dealt with the specific comments.

In my opinion this paper will be suitable for publication in Biogeosciences after moderate revision.

**Major arguments**

The model used in this study is only forced using one (extreme) climate scenario and there is a lot of (unquantified) uncertainty in the results. The authors do already state in the discussion that this uncertainty exists and could not be solved due to computational restrictions and because there currently are no different climate scenarios at the high resolution of this study available. Despite this, the study is still a relevant and necessary first step. However, I do feel it is very important to mention the uncertainty of this study more clearly in both the conclusion and the summary to avoid sensationalizing the results. If I were to read only the title, summary, and conclusion of this paper, I would not know this paper only describes general trends. Especially in the summary that states this region **will** be completely covered by forests at the end of the 21st century.

The conclusion (679) and summary (741) already clearly state that this is a high-emission scenario. Some of the uncertainty discussion has been moved to the conclusion (689-696).

In my opinion both the methodology and introduction require some additions to make it easier for people to find the relevant information and to clearly express the relevance and necessity of this study:

The modelling methodology is in my eyes poorly described and rather wordy. It is unclear what model is used and where a detailed description of this model can be found. I had to consult additional literature to try and figure out what model was used (Gustafson et al., 2021; Smith et al., 2001; Miller and Smith, 2012; Smith et al., 2014) and I am still not sure, because Gustafson at al. (2021) only mentions LPJ-GUESS v4.0. This could be remedied by adding a schematic overview of the model highlighting which parts of the model are new (reindeer module and some PFTs) and where the detailed descriptions for the old parts can be found.

We have clarified what model was used and where it is described in detail (128-131). We have not added a schematic overview, but do describe clearly the new features and parameterizations you mention.

Additionally, it is unclear which data was used to calibrate which parameters of the model. For example, lines 143-147 on page 3 mention test runs and fine-tuning to get a better match against distribution maps from observations, but it does not mention what maps. I feel adding a (supplementary) table of which data sets were used to calibrate which parameters (covering what time period) would solve this problem and prevent people having to comb through the entire paper to figure out what was done.

As also pointed out Referee #1 and #2, we admit that it is not clearly expressed. We have revised the text (156-159), and added a new section in S2 (S27-36). In Table S2 we added the parameters that were not changed according to Gustafsson and added a column so that there is "default 4.1", "Gustafsson et al. (2021)" and "used value" (Tab S2). Except for this there was no calibration done, and we don't think any table more than Table S2 will be needed.

In the introduction the paper is vague on what the consequences of a shift in vegetation composition might be and how reindeer grazing might affect and be affected by this shift. It just mentions increasing pressure on both ecosystems, holding species of great ecological, biological, and societal significance, and societies in the area. It seems to me that the consequences of such a shift in vegetation due to climate change are the most important reason for doing this research. I feel it would be beneficial to be more specific or perhaps add some examples to get the message across better. Stark et al. (2023) gives a lot of information about effects of reindeer on the ecosystem. It is also worth mentioning how culturally important reindeer husbandry is for the indigenous people of the region.

We now explain consequences of climate-change related shifts in vegetation composition to show that this research is important using the Stark reference, and mention the cultural importance for the Sami (68-72). We now also more clearly express the task, objective and hypothesis of the study in the end of the introduction (100-103).

Building further on this point: The summary and conclusion mention more about the consequences than the introduction does. The conclusion for example mentions implications for recreation when this is not mentioned anywhere else. These extra points should be moved to or already be mentioned in the introduction.

Tourism is now mentioned in the introduction (103).

I would also like to see more discussion about processes and effects that were not considered in the model. The following points being most important:

Currently only one sentence in the paper (line 607) mentions seed dispersal capacity (and fire disturbance) as factors that may restrict vegetation expansion, particularly for predicted shrub expansion on non-shrub tundra. However, as mentioned by Gustafson et al. (2021) dispersal limitations are likely to cause lags in range shifts on larger spatial scales (Rees et al., 2020; Brown et al., 2018). Models that do account for seed dispersal limitations generally predict slower latitudinal tree migration than models driven solely by climate like LPJ-GUESS (Epstein et al., 2007). This warrants further explanation in the paper.

We have extended the discussion with possible consequences of not accounting for seed dispersal rate (655-659).

This paper discusses the direct effects reindeer grazing, browsing, and trampling might have on the vegetation and how the change in vegetation due to climate change might affect the food supply for the reindeer, but it does not discuss the effect reindeer might have on the climate and thus indirectly on the vegetation as well. Recently Holmgren et al. (2023) found that a high amount of reindeer summer trampling in low peatland areas may result in increased summer warming and decreased winter cooling enhancing permafrost degradation in these areas. On the other hand, in higher areas intense browsing and nutrient addition from reindeer may mitigate some climate warming effects (Macias-Fauria et al., 2020; Malhi et al., 2022). This uncertainty should be added to the discussion.

We have added a discussion about possible climate feedback from reindeer grazing (604-606).

Furthermore, it should be noted that the reindeer exclosure sites that are used to validate the reindeer grazing module likely do not represent the natural situation without grazing (Stark et al., 2023).

We have added a discussion of both simulated and observed uncertainty of this aspect (644-649).

Soil vertical and horizontal movement caused by frost, and amelioration of such effects in the warmer future climate are not accounted for in the LPJ-GUESS model. These processes could affect survival and competition among the plant functional types, especially in the seedling stage when plants are most vulnerable to mechanical disturbance (Gustafson et al., 2021). This should also be added to the discussion, if relevant on high resolution larger scale.

There are a lot of simplifications when it comes to considering processes that affect the establishment of new plants; the mentioned soil processes, seed predation, plant browsing and killing by smaller animals like rodents and hares, deceases, snow damage, moose, etcetera. We now mention this in general terms (663-666), but an extended discussion of all these aspects would take too much space.

**Minor arguments**
1. Page 3, lines 104-105: Why was this study area chosen? The summary, introduction and title make it feel like the entire Fennoscandian area will be studied, when it is not. I feel like some explanation is necessary here.

In the introduction we clearly state that we address the "Fennoscandian boreal and Oroarctic region" (55, 85, 96). That the study is also restricted to the area used for reindeer herding is explained in M&M (107-108), but we have now added this information also to the introduction (96). The eastern border in Finland, where a small part that could have been included is outside the assessed area, is due to the fact that we for computational reasons wanted to restrict the size of the complete 1985-2100 NetCDF climate files to less than 32 GB per climate variable, this information is now clearly stated in section 2.3 (192-195).

2. Page 5, line 128: The return time of patch destroying disturbances (e.g., devastating pests or windstorms) is set to 150 years. What is this based on? Does this also include fires? This requires some explanation.

Fire is simulated separately, we have added information on that (142-143). 150 years is the default return time for arctic simulations in LPJ-GUESS 4.1, we have added that information as well (141-142).

3. Page 6, lines 158-159: Barthelemy et al. (2018) says that nitrogen in the form of urea can easily be taken up by plants directly (as in without transforming it, not as in how quickly). The way it is currently formulated in this paper makes it feel as if the assumption is that the nitrogen is immediately (time) taken up by the plants. Barthelemy et al. (2018) does not seem to state how quickly this happens other than that it happens 'rapidly'. Urine is also a very local phenomenon, but for the model it is taken up by plants in the entire patch.

We agree that "directly" is a misleading choice of word and have changed to "rapidly" both here (180) and in the supplement (S72).

4. Page 7/8, section 2.4: Is the nitrogen data at 12 km resolution detailed enough for this study with a resolution of 3 km? This is not mentioned anywhere yet.

We have done a sensitivity test using the more coarsely resolved 0.5° Ndep data and compared it to simulations using the high-resolution data, and found very small differences in simulated vegetation composition. We have added this information but do not provide specific results (252-254).

5. Page 15/16, section 3.2.2: I wonder what the effect is of leaving out the wetland areas in the calculation of the Shannon Diversity Index on the comparison between the different time periods. Specifically in Muddus, which has a lot of wetland area in the simulated data, but barely any in the satellite data. This should be mentioned.

We explain in M&M that we exclude cells that do not have a dynamic vegetation class like water and wetlands (312-313), but we have added this information also to the table text of Table 3 (467). We have also added a comment regarding these results (456-458).

6. Page 22, line 556: I feel it would be better to mention just how much larger the reindeer herd size is in summer (60% larger after calving and before slaughter, Definitions - Sámi Parliament (sametinget.se)), because this could result in a significant underestimation of the grazing.

We can't see where the number 60% comes from, according to https://sametinget.se/statistik/renslakt (in Swedish) the number of slaughtered animals is 40 000 – 75 000 reindeers, which rather means that the summer population is about 25% larger. The animals that are slaughtered are also often calves that eat less. We already acknowledge that there is an underestimation in the current formulation. We have not revised.

7. Supplement page 4 (S3), line 38: Please add and explanation as to why the Swedish reindeer population is (assumed to be) representative for the entire study area.

We have added the information that we will use Sweden as an example for verification (S52). We write in the main text that the focus of the study will be on Swedish ecosystems (108).

8. Supplement page 4 (S3), lines 53-59: This does not explain why 35% was used. Ferraro et al. (2022) assumes that 38% of the daily nitrogen consumed by reindeer is assimilated into its body mass and 62% is defecated (not all as urine), instead of the other way around. McEwan and Whitehead (1970) show that nitrogen divers a lot depending on the age and sex of the reindeer and on the season. Nitrogen retention during the second growth phase (14-24 months) was on average 35% in reindeer (page 909, table IV). The remaining 65% was presumed to be urine. I am presuming the 35% used in this study is based on this, but I am not entirely convinced this is representative for the entire reindeer population. Please explain in more detail.

It is not the other way around, entering the harvest pool is the same as being assimilated in body mass, e.g. being removed from circulation in the ecosystem. To go into detail of the proportion of different growth phases would be overly complicated for this simple model, but we have added a comment (S75).

**Minor issues:**
Main article
Page 1, lines 30-32: "Simulated … grazing ground." This sentence is unclear and needs to be rewritten.

For clarity, we have split the sentence into two (30-32).

Page 2, line 37: Reference?
A reference has been added (38).

Page 3, line 87: "have" should be "has".
"Data" is always plural (datum is the singular form), we have not revised.

Page 3, line 95: Perhaps add a more recent reference for the "state-of-the-art DMV" here like Gustafson et al. (2021). 2014 is not state-of-the-art.
Lindeskog et al. 2021 has been added (94).

Page 6, line 175: "(see below)" What does this reference? There is nothing below.
We have removed this reference as it is not needed (from 198).

Page 7, line 182: "two" methods" are announced here, but only one of those methods is explained in the same paragraph and the other in the next paragraph.
We have placed a colon after "using two methods" (205) and then start a new paragraph, so that each method has a separate paragraph (207-211 and 213-223).

Page 8, line 246: "also" before "converted" should be left out.
We have revised the entire sentence (276-279).

Page 13, line 363: "were" should be "was".
We have revised accordingly (406).

Page 21, lines 524-525: "which is not far from the current trajectory" reference?
We have revised the entire sentence and added references (562-565).

Page 22, line 554: either "since" or "as" needs to be left out.
"since" has been removed (597).

Page 29, line 805: This references the preprint of this paper, not the published version.
It has been fixed (881-883).

*Supplement*
Supplement page 4, line 34: "reduce" should be "reduces".
We have revised accordingly (48).

Supplement page 4, line 39-40: add the specific webpage where the numbers used can be found.
The webpage we are referring to specifically provides these numbers, though it is in Swedish.

Supplement page 4, line 40: "280 00" should be "280 000".
The missing zero has been added (54).

Supplement page 4, line 41: "eat" should be "eats".
We have revised accordingly (56).

Supplement page 4, line 41: I think "path" is supposed to be "patch" here.
Correct, we have revised (55).

Supplement page 4, line 46: missing "the" between "large" and "fraction".
We have revised accordingly (61).

Supplement page 4, line 47: add "is" between "consumed" and "relative" and perhaps add "where" between "and" and "herbivore_int" to make the sentence clearer.
We have revised accordingly (62).

Supplement page 4, line 55: missing "a" between "is" and "functionality".
We have revised accordingly (71).

Supplement page 4, line 55-56: add a reference for the assumption that N leaving the herbivore as urine is directly taken up by the plants.
We have added a reference (72).

Supplement page 7, line 112: The Bartalev et al. (2003) reference is missing in the reference list.
Supplement page 8, line 119: Babst et al. (2014) is not referenced in the text.
By mistake Babst et al. (2014) was inserted in the reference list instead of Bartalev et al. (2003). The mistake has been corrected (S143-145).

Supplement page 18-19: What do the colours in this figure mean?
We have added an explanation of the colours (S229-231).

**References**
Barthelemy, H., Stark, S., Michelsen, A., & Olofsson, J. (2017). Urine is an important nitrogen source for plants irrespective of vegetation composition in an Arctic tundra: Insights from a 15 N-enriched urea tracer experiment. Journal of Ecology, 106(1), 367–378. https://doi.org/10.1111/1365-2745.12820

Brown, C. M., Dufour-Tremblay, G., Jameson, R. G., Mamet, S. D., Trant, A. J., Walker, X. J., Boudreau, S., Harper, K. A., Henry, G. H. R., Hermanutz, L., Hofgaard, A., Исаева, Л. Г., Kershaw, G. P., & Johnstone, J. F. (2018). Reproduction as a bottleneck to treeline advance across the circumarctic forest tundra ecotone. Ecography, 42(1), 137–147. https://doi.org/10.1111/ecog.03733

Epstein, H. E., Qin, Y., Kaplan, J. O., & Lischke, H. (2007). Simulating future changes in Arctic and Subarctic vegetation. Computing in Science and Engineering, 9(4), 12–23. https://doi.org/10.1109/mcse.2007.84

Ferraro, K. M., Schmitz, O. J., & McCary, M. A. (2021). Effects of ungulate density and sociality on landscape heterogeneity: a mechanistic modeling approach. Ecography, 2022(2). https://doi.org/10.1111/ecog.06039

Gustafson, A. F., Gustafson, A. F., Björk, R. G., Olin, S., Smith, B., Björk, R. G., Olin, S., Smith, B., & Smith, B. (2021). Nitrogen restricts future sub-arctic treeline advance in an individual-based dynamic vegetation model. Biogeosciences, 18(23), 6329–6347. https://doi.org/10.5194/bg-18-6329-2021

Holmgren, M., Groten, F., Carracedo, M. R., Vink, S., & Limpens, J. (2023). Rewilding risks for Peatland permafrost. Ecosystems. https://doi.org/10.1007/s10021-023-00865-x

Macias-Fauria, M., Jepson, P., Zimov, N., & Malhi, Y. (2020). Pleistocene Arctic megafaunal ecological engineering as a natural climate solution? Philosophical Transactions of the Royal Society B, 375(1794), 20190122. https://doi.org/10.1098/rstb.2019.0122

Malhi, Y., Lander, T. A., Roux, E. L., Stevens, N., Macias-Fauria, M., Wedding, L. M., Girardin, C., Kristensen, J. A., Sandom, C. J., Evans, T., Svenning, J., & Canney, S. M. (2022). The role of large wild animals in climate change mitigation and adaptation. Current Biology, 32(4), R181–R196. https://doi.org/10.1016/j.cub.2022.01.041

McEwan, E. H., & Whitehead, P. E. (1970). Seasonal changes in the energy and nitrogen intake in Reindeer and Caribou. Canadian Journal of Zoology, 48(5), 905–913. https://doi.org/10.1139/z70-164

Miller, P. A., & Smith, B. (2012). Modelling Tundra vegetation response to recent Arctic warming. AMBIO: A Journal of the Human Environment, 41(S3), 281–291. https://doi.org/10.1007/s13280-012-0306-1

Rees, G., Hofgaard, A., Boudreau, S., Cairns, D. M., Harper, K. A., Mamet, S. D., Mathisen, I. E., Swirad, Z., & Tutubalina, O. (2020). Is subarctic forest advance able to keep pace with climate change? Global Change Biology, 26(7), 3965–3977. https://doi.org/10.1111/gcb.15113

Smith, B., Prentice, I. C., & Sykes, M. T. (2001). Representation of vegetation dynamics in the modelling of terrestrial ecosystems: comparing two contrasting approaches within European climate space. Global Ecology and Biogeography, 10(6), 621–637. https://doi.org/10.1046/j.1466-822x.2001.t01-1-00256.x

Smith, B., Wårlind, D., Arneth, A., Hickler, T., Leadley, P., Siltberg, J., & Zaehle, S. (2014). Implications of incorporating N cycling and N limitations on primary production in an individual-based dynamic vegetation model. Biogeosciences, 11(7), 2027–2054. https://doi.org/10.5194/bg-11-2027-2014

Stark, S., Horstkotte, T., Kumpula, J., Olofsson, J., Tømmervik, H., & Turunen, M. (2023). The ecosystem effects of reindeer (Rangifer tarandus) in northern Fennoscandia: past, present and future. Perspectives in Plant Ecology, Evolution and Systematics, 58, 125716. https://doi.org/10.1016/j.ppees.2022.125716

**RC2**: 'Comment on bg-2023-148', Anonymous Referee #2, 21 Nov 2023
**General comments**
**This study focuses on a subject of great scientific importance: changes in vegetation at high latitudes. The work presented is of high quality and based on state-of-the-art data and methods. The figures and the introduction and discussion sections are of good quality and in good shape for publication. The M&M and results sections could benefit from some changes in content and structure (see following paragraphs).**
Thank you for positive words and many useful comments, we have tried to improve the M&A and discussion sections following your more specific comments.

**Specific comments**
**Introduction**
The introduction is clear, synthetic, and overall presents a sufficient number of scientific references. It describes the processes taking place in high-latitude regions, in particular the impacts of climate change and reindeer husbandry on the different ecosystems (boreal forest, tree-line and tundra). It presents the implication of vegetation dynamics modeling (DVM) and the added value enabled by scenario climate data at a 3km scale.

One introduction paragraph (line 54-61) should be moved to the Material and Method section, as it describes the study area and interrupts the flow developed in the introduction.
The paragraph has been moved (from 55 to 108-115)

In addition, it would be interesting to present the initial hypotheses of the study, particularly with regard to the description made of the causes (societal and climatic) of change in plant communities.
As also the other reviewers have pointed out that task, objective and hypothesis should be clearly stated, we will revise the last section of the introduction. We have added a clarification of the task (91-92) and specified the objectives (100-103).

Finally, it would be interesting to mention the LPJ-Guess model in the paragraph on DVM, in order to highlight its field of application.
We now mention LPJ-GUESS in this paragraph (81-84).

**M&M**
This section is well presented and provides a clear understanding of the data and studies used to complete this scientific production. More detail could be provided on certain paragraphs (see specific comments). More specifically, the choices made regarding data acquisition, interpolation type and parameter tuning must be sufficiently justified or discussed.

A comparative analysis of different climate (RCP) or socio-economic (SSP) scenarios would have been desirable. Thus, the authors of the study are invited to explain this choice of scenario.
With the addition to the last paragraph of the introduction suggested above, we don't think it would be needed to elaborate on the choice of scenario in M&M.

**Results**
This section provides well the overall results of the study with different type of figures. This section is quite dense and would benefit from structural reshaping, including topic sentences introducing most paragraphs.
We have not revised the general structure of the Results, or added introductory sentences to the sections as we think it would be a waste of space. To help the readers through the Results section we have instead provided more specific sub headings (329, 368-369, 398, 428, 494, 511).

**Discussion & Conclusion**
I truly enjoyed reading the Discussion section which adequately explain the results and does not hide the limits of the method. I have very few comments on this section.
Thank you.

**Technical corrections**
38 « biological and societal significance » What do you mean by biological significance?
We have removed "biological" (37).

82 « This information does not capture all local variation, especially in areas of complex terrain where altitudinal differences can be strongly underestimated » This sentence would require a reference.
We have added the Lind et al. (2020) reference also to this sentence (80).

91 « km-scale » Would be more rigorous to refer to 3km-scale throughout the text
The idea behind the term "km-scale" in climate modelling is to distinguish km-scale (implying 1-4 km) from coarser resolutions (>10 km). In other words, "km-scale" stands for the order of magnitude of 1 km (i.e. < 5 km), rather than the specific value of 1 km. We will still use the more general "km-scale" in more general contexts such as in the title and in line 93, and we will explain what the term means in connection to that line (102-105).

110 « The study area (shown as altitude from sea level (dark green) to 2000 m a.s.l. (yellow), the six focus "hotspot" areas (shaded squares and black text, see Figure S1 for detailed maps) » Instead of adding the description of symbols and colors in the text, use a legend directly in the figure with color (even for altitude) and symbol. What's more, the names don't appear clearly.
We have added an altitude legend and revised the figure text, lines and symbols to be more legible (Fig 1).

127 « disturbances representing e.g. devastating pests or wind storms, occur randomly in each patch (return time set to 150 years in the presented simulations) » In natural/semi-natural ecosystem, disturbance does not occur randomly. Some ecosystems are more fire-prone (e.g. fire) depending on the season. You might want to discuss the "150yr return time at random" in the discussion, especially considering its huge significant impact in vegetation dynamics
We now discuss uncertainty related to disturbance return time (659-663).

139 « For fractions of land classified as peatland » Include peatland location in Fig. 1.a.
The peatland areas are too small and scattered to be shown in the overview of Figure 1a. The areas are shown in Figures 2, 4 and 6 as the peatland area is not dynamical but prescribed from this "PEATMAP". But due to the scale they can only be detected in Figure 6.

145 « A fine-tuning of some of the model's parameters was therefore done to get a better match against distribution maps from observations » Your parameter changes seem significant, especially for growing degrees days. Could you explain your choices in more detail? Does LPJ-GUESS present a poor calibration of this PFT in general or for this geographic zone in particular?
As also pointed out Referee #1 and Leanne, we admit that it is not clearly expressed. We have revised the text (156-159), and added a new section in S2 (S27-36). In Table S2 we added the parameters that were not changed according to Gustafsson and added a column so that there is "default 4.1", "Gustafsson et al. (2021)" and "used value" (Tab S2).

150 « Grazing/browsing was simulated by removing a fraction of leaf biomass. » Provide a reference to support the assumption that grazing/browsing only/mainly affects leaf biomass. Some could argue that plant survival or growth could also be directly affected.
LPJ-GUESS has only stem, leaf and fine-root compartments, grass PFTs only leaf and fine-roots. We now explain this in section 2.2.2 (169-172). We have not found any study of what proportion of the consumed biomass that is twigs/branches. We now acknowledge that we may underestimate the effect of reindeer browsing in the M&M (171) and in the discussion (644-647) We have also added a comment that there are rather small differences in simulations with or without reindeer presence in section 3.3.1 (500).

200 « The RCP8.5 scenario used was the first dataset produced at this high resolution for the entire region » The first and only one? Please indicate why using this particular scenario and not propose a comparative analysis with Shared Socioeconomic Pathway to get a nuanced vision of vegetation dynamics in the future.
As stated above, we now state already in the introduction that there indeed really was only this scenario available at this resolution (100).

210 « 5 arc seconds (10 km) » Please use consistent unit for resolution (km preferred)
This is not straight forward, as at these high latitudes the distance for a degree unit is 2-3 times larger in S-N direction than in W-E, depending on latitude (https://www.nhc.noaa.gov/gccalc.shtml). To be more consistent we have used km as main unit and for data originally in degree grid we have present the resolution in km in S-N and W-E in parenthesis (154-155, 235, 251). At this specific place we also made a mistake, "seconds" should have been "minutes", the unit is now, however, given as decimal degrees (5/60 = 0.0833) for consistency (235).

211 « had been interpolated to 3 km resolution » What type of interpolation? "interpolated" is not the appropriate expression here as they were just taken from the relevant 10 km data cell. We have just skipped the expression after the parenthesis (236).

225 « downscale the 50 km data for the 1900-1986 period. After 2051, the 0.5° resolution Lamarque et al. (2011) dataset was used, which is standard for LPJ-GUESS. » How the final resolution of 3km is reached?
The Ndep data were simply at a coarser resolution. We are aware that this could be a potential problem, but we have done a sensitivity test using the coarser 0.5° Ndep data and compared it to simulations using the high-resolution data, finding very small differences in simulated vegetation composition. We have added this information but do not provide specific results (252-254).

226 « 0.5° » require consistent unit
This is not straight forward, as at these high latitudes the distance for a degree unit is 2-3 times larger in S-N direction than in W-E, depending on latitude (https://www.nhc.noaa.gov/gccalc.shtml). To be more consistent we have used km as main unit and for data originally in degree grid we have present the resolution in km in S-N and W-E in parenthesis (154-155, 235, 251).

267 « to convert modelled total biomass to above ground biomass we assumed a factor of 0.85 based on earlier estimates » Root:shoot ratios could be very variable for foret tree species. For instance, in (Huttunen et al. 2013) they estimated that for silver birch seedlings, the ratio between BGB and AGB could vary from 0.35 to 1.2 under different conditions and would be around 0.55 without treatment. Even though you consider here adult trees, the approximation of 0.85 should be discussed.
Huttunen, Liisa & Ayres, Matthew & Niemelä, Pekka & Heiska, Susanne & Tegelberg, Riitta & Rousi, Matti & Kellomäki, Seppo. (2013). Interactive effects of defoliation and climate change on compensatory growth of silver birch seedlings. Silva Fennica. 47. e964. 10.14214/sf.964.
We have added the Huttunen et al. (2013) reference and added an explanation why we used the higher value of Johansson (2007) (298-300).

299 « Simulated_LAI = 0.78 × SURFEX_LAI − 1.99, r2 = 0.59 » You might consider providing the results with an intercept of 0. In addition, even if an r² of 0.59 can be considered satisfactory, you should explain where and why the simulated LAI differs from the SURFEX data.
As there is such a large intercept (-1.99), we don't think it would be useful to show statistics for a regression through zero. We now comment on this large intercept and give reasons for the big difference (330-335).

340 « Figure 3» Compared to the high quality of the other figures, this one could be reshaped and made more clear
RC2's comment is not specific. To distinguish the different datasets we have added a pattern to the simulated bars, we have also made a minor revision of the lines (Fig 3).

381 « As the classification is based on LAI, bare rock was set for LAI 0.01- 0.001 and permanent snow/ice < 0.001, indicating that plants have the potential to grow there » Unclear and might be put in M&M section

Instead of the details given between the commas in the sentence, we now refer to supplement S5 where it is described in detail (421-422).

385 « Figure 5 » This figure is really interesting and well designed, congrats. One thing, to be more rigorous, the 3 rows should me made from equivalent latitude band sample ( not 0.2, 0.15 and 0.1 °)
Thank you for the appreciation. Again, the bands should not be equally wide as the number of gridcells per latitude degree is different. The previous areas of the bands were 10000, 8500 and 7300 km$^2$ from north to south. We have now revised the figure varying the width of the bands so that they each has an area of 9000 km$^2$ and consistent density of dots (Fig 5). There was no change in the general pattern of the figures (see the marked-up version of the manuscript).

442 « It should be kept in mind that the data obtained from the Analysis Portal relies on what has been reported by a large community of public and professional naturalists, which means that biases can exist e.g., depending on the specific biological interests of rapporteurs visiting the different areas. » this sentence might be preferably put in Discussion section and more developed. Indeed, this section aim at providing an overview of selected hotspot diversity but in my opinion the method for this purpose should be more discussed.
We have move this statement to the discussion (from 483 to 573-576). We think that discussion of the details of methods for collecting this type of data would take too much focus from the main aspects of the article.

478 « Figure 8 » The colour legend might be redefined to better show region with low and high change of consumption. In addition, it could be interesting to plot the relative change (percentage) instead of the absolute one which poorly reflects effect on vegetation.
We have changed the colour scheme so that it starts and ends at darker colours (Fig 8). To show relative change is not possible as many grid cells are at zero in present period.

501 « dramatic » This term has a negative connotation, but the consequences of such a change in vegetation are complex and not entirely negative, depending on the process under consideration.
We have changed to a more neutral word: "extensive" (541).

544 « This is because many of the alpine species in the hotspots areas that are not listed today will be threatened as warming continues. » This need a literature reference.
We have added a reference (589).

**Editor comments**
We have tested the figures in the colour blindness simulator. There is a lot of categories to show in the figures and we have used standard colour schemes that we think work OK (though not perfect in all variants). Figure 7 was somewhat difficult, to make it easier to interpret we have reworked the legend so that the PFTs are now in the same order in the legend as in the figure, which should help (see the marked-up version).

There have been no changes to numbers in the tables and figures except to the additional data in Table S2. The change to Figure 5 removed or added a few dots in the different sub figures but there was no change in the general pattern (see the marked-up version).

Some minor corrections have been made not suggested by reviewers (see the marked-up version).

---

## Author Response (AR2)

The revised manuscript was not sent for review

The editor found some web-links that did not worked. We have checked all links in the manuscript and fixed the ones that had been outdated. Some links had been broken when converting to PDF due to the syllabication of the Word-template, these have also been fixed now.

Outdated fixed:

Page 9: https://artfakta.se/artinformation/lists

Page 11: https://www.lantmateriet.se/en/

Page 18: https://forobs.jrc.ec.europa.eu/glc2000/data

Broken fixed:

Page 6: https://data.apps.fao.org/map/catalog/static/search?keyword=DSMW

Page 9: https://www.analysisportal.se/

Page 17: https://zenodo.org/record/4501636#.ZBqvOXbMJaQ

Page 17: https://previous.iiasa.ac.at/web/home/research/researchPrograms/air/ECLIPSEv6b.html

Page 18: https://land.copernicus.eu/pan-european/corine-land-cover/clc2018

After checking the guidelines, all references to figures have been changed from "Figure" to "Fig.".

"Competing interests" declaration added